# Latin Americans show wide-spread Converso ancestry and imprint of local Native ancestry on physical appearance

Juan-Camilo Chacón-Duque (ORCID) et al.[#]

Historical records and genetic analyses indicate that Latin Americans trace their ancestry mainly to the intermixing (admixture) of Native Americans, Europeans and Sub-Saharan Africans. Using novel haplotype-based methods, here we infer sub-continental ancestry in over 6,500 Latin Americans and evaluate the impact of regional ancestry variation on physical appearance. We find that Native American ancestry components in Latin Americans correspond geographically to the present-day genetic structure of Native groups, and that sources of non-Native ancestry, and admixture timings, match documented migratory flows. We also detect South/East Mediterranean ancestry across Latin America, probably stemming mostly from the clandestine colonial migration of Christian converts of non-European origin (Conversos). Furthermore, we find that ancestry related to highland (Central Andean) versus lowland (Mapuche) Natives is associated with variation in facial features, particularly nose morphology, and detect significant differences in allele frequencies between these groups at loci previously associated with nose morphology in this sample.

[#]A full list of authors and their affiliations appears at the end of the paper.

The history of Latin America has involved extensive admixture between Native Americans and people arriving from other continents, particularly Europe and Africa[1–3]. Most genetic studies carried out to date have examined this process mainly in relation to variation in overall Native American, European and Sub-Saharan African ancestry across regions and between individuals[2–4]; with small and geographically-restricted East Asian ancestry also reported[5–7]. In addition, some genetic analyses have sought to detect regional ancestry within the three major continental components, i.e. the sub-continental origins for individuals having contributed to admixture in Latin America. For instance, mtDNA and Y-chromosome data suggest that historical admixture in North West Colombia involved local Native women, and that some immigrant men carried haplogroups common in Jewish populations[8]. The inference that historical admixture of Latin Americans in specific regions involved Natives with a relatively close genetic affinity to those currently living in the same areas was subsequently supported using genome-wide autosomal data[9,10]. Recent genome-wide SNP studies (GWAS), partly implementing haplotype-based analyses, have further expanded the notion that the demographic shifts of the last few generations have not entirely erased signals of historical population structure in Latin America[6,11–15]. A finer characterization of the admixture history of Latin America would benefit from a more extensive sampling across the region, as well as from further methodological improvements (including fully haplotype-based analyses and improved modelling approaches) and a wider survey of reference population samples (from areas potentially contributing to Latin American admixture).

The broad significance of characterizing these fine-grained patterns of human genetic diversity in Latin America is emphasized by the realization that geographically-restricted genetic variation is potentially a key component of the genetic architecture of common human phenotypes, including disease[16]. Furthermore, studies of regional human genome diversity, and its bearing on phenotypic variation, have so far been strongly biased towards European-derived populations[17]. The study of populations with non-European ancestry is essential if we are to obtain a more complete picture of human diversity. Latin America represents an advantageous setting in which to examine regional genetic variation and its bearing on human phenotypic diversity[18], considering that the extensive admixture resulted in a marked genetic and phenotypic heterogeneity[2,3,19]. Relative to disease phenotypes, the genetics of physical appearance can be viewed as a model setting with distinct advantages for analyzing patterns of genetic and phenotypic variation. Many physical features are relatively simple to evaluate, show substantial geographic diversity and are highly heritable. We have previously shown that variation at a range of physical features correlates with continental ancestry in Latin Americans[19] and have identified genetic variants with specific effects for a number of features[20–22].

Here we apply fully haplotype-based methods that have been shown to provide higher resolution than allele-based approaches for examining patterns of human population sub-structure[23], for example recently enabling a fine-grained analysis of the population structure and demographic history of the British Isles[24]. We present a novel model-based technique for ancestry estimation with a substantial increase in accuracy compared to the technique used in the aforementioned study. We applied this technique to the largest Latin American sample available to date, and an extensive set of reference population samples, in order to delineate patterns of sub-continental genetic diversity across Latin America. Our results demonstrate a striking geographical correspondence between Native ancestry components in Latin Americans and the genetic structure of present-day Native groups. We also match non-Native ancestry components to precise regions of Europe at a sub-country level and unearth ancestry related to present-day groups from the East/South

Mediterranean, Africa and East Asia. We infer the timings of these genetic contributions and relate them to historically-attested migrations, for example providing compelling new evidence of widespread ancestry from undocumented migrants during the colonial era. We further show how differences in Native and European sub-continental ancestry components are associated with variation in physical appearance traits in Latin Americans, highlighting the impact of regional genetic variation on human phenotypic diversity.

## Results

**Overview of the data.** We examined data for over 500,000 autosomal SNPs typed in more than 6,500 individuals born in Brazil, Chile, Colombia, Mexico and Peru (denoted the CANDELA sample, Supplementary Fig. 1). To infer ancestry in this sample, we also collated data for 2,359 individuals from 117 reference populations (including 430 newly genotyped individuals from 42 populations) representing five major bio-geographic regions: Native Americans; Europeans; East/South Mediterraneans; Sub-Saharan Africans and East Asians (Fig. 1a, Supplementary Table 1, Supplementary Fig. 2). Analysis of these data using the allele-based approach ADMIXTURE[25] shows major limitations for characterising sub-continental ancestry (Supplementary Note 1), similar to what has been observed with other datasets[23,26]. We therefore performed fully haplotype-based analyses. We first grouped the reference population individuals into 56 homogeneous clusters based on patterns of haplotype sharing, primarily using the program fineSTRUCTURE[23], followed by secondary refinements (see Methods, Supplementary Tables 2 and 3). We inferred the proportion of the genome in each CANDELA individual that is most closely related to each of these 56 surrogate clusters, using a novel approach we term SOURCEFIND (see Methods). In contrast to another haplotype-based approach that implements a Non-Negative least squares (NNLS) regression[24,27], SOURCEFIND uses a Bayesian model that eliminates contributions that cannot be reliably distinguished from background noise. Simulations show that SOURCEFIND has greater accuracy than NNLS (Supplementary Note 2). For ease of visualization, we collapsed the ancestry components inferred from the 56 surrogate clusters into 35 groups, based on the genetic relatedness of the clusters (Supplementary Fig. 3). Average continental and sub-continental ancestries from SOURCEFIND and ADMIXTURE are provided in Supplementary Note 3.

**Patterns of Native American ancestry in the CANDELA dataset.** Anthropological studies indicate that Pre-Columbian Native population density varied greatly across the Americas, impacting on the extent of Native American ancestry observed across Latin America[2]. Native ancestry in the CANDELA sample varies considerably between countries, and we also observe a marked geographic differentiation in sub-continental Native ancestry within each country, with a strong correspondence with the genetic structure of the Native American reference groups (Figs. 1 and 2). Allele-based analyses have previously documented that broad patterns of Native American population structure are detectable in admixed Latin Americans[10,14]. Our haplotype-based analyses significantly extend these results by enabling the inference of 25 Native American ancestry components across Latin America (Supplementary Fig. 3), which we combined into 16 components for visualization (Figs. 1b and 2a). In Mexicans we find a predominant Nahua sub-component (most prevalent across northern and central Mexico) and two smaller sub-components, one related to Natives of south Mexico and another to Mayans (seen mainly in Mexicans from Yucatan), similar to previous reports[14,28]. In Peruvians we observe a predominant Quechua component (in central Peru), a sub-component related to Andean-Piedmont Natives (concentrating in Northern Peru) and

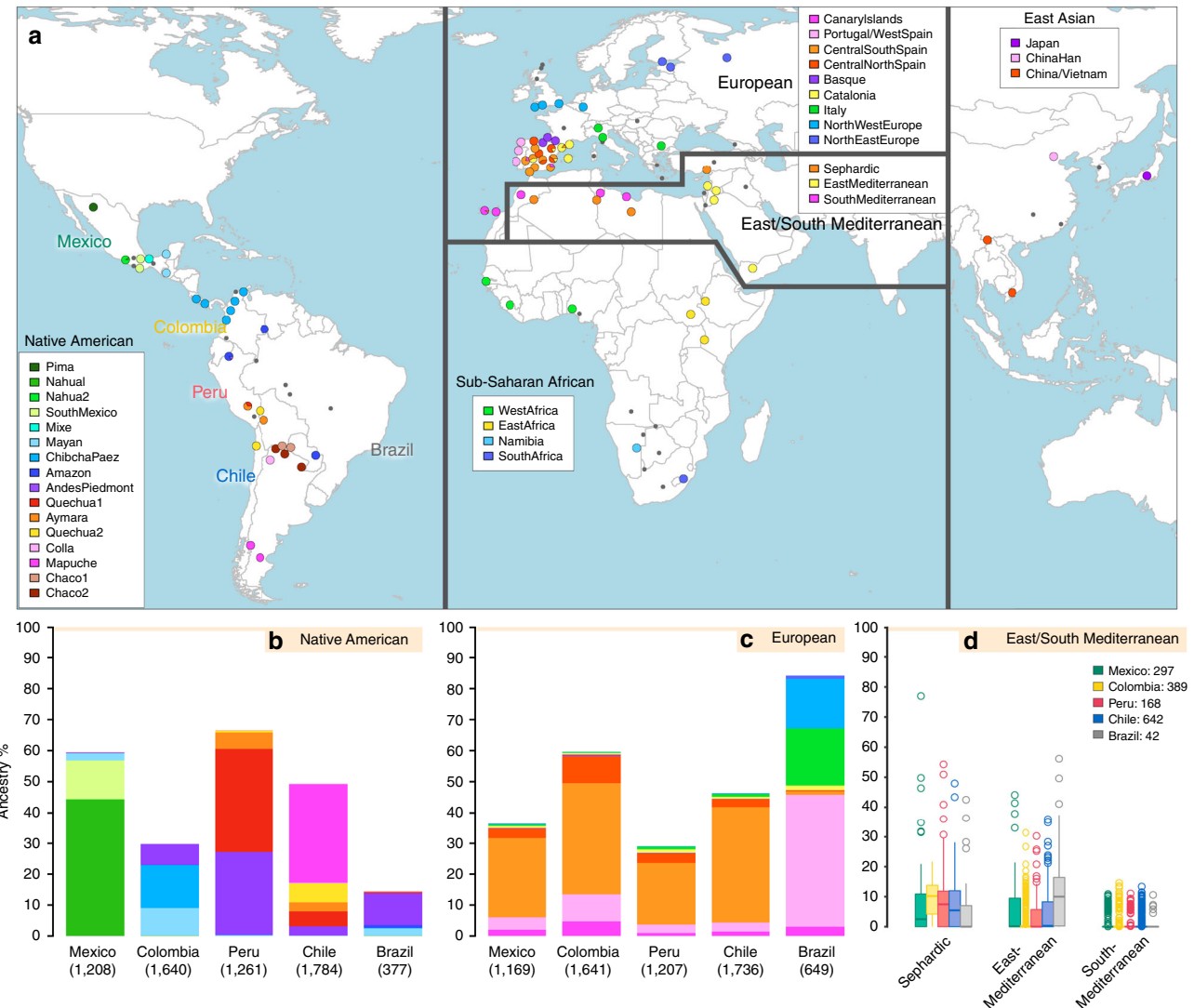

**Fig. 1** Reference population samples and SOURCEFIND ancestry estimates for the five Latin American countries examined. **a** Colored pies and grey dots indicate the approximate geographic location of the 117 reference population samples studied. These samples have been subdivided on the world map into five major bio-geographic regions: Native Americans (38 populations), Europeans (42 populations), East/South Mediterraneans (15 populations), Sub-Saharan Africans (15 populations) and East Asians (7 populations). The coloring of pies represents the proportion of individuals from that population included in one of the 35 reference groups defined using fineSTRUCTURE (these groups are listed in the color-coded insets for each region; Supplementary Fig. 2). The small dark grey dots indicate reference populations not inferred to contribute ancestry to the CANDELA sample. **b**–**d** refer to the CANDELA dataset. **b**, **c** show, respectively, the average estimated proportion of sub-continental Native American and European ancestry components in individuals with >5% total Native American or European ancestry in each country sampled; the stacked bars are color-coded as for the reference population groups shown in the insets of (**a**). **d** shows boxplots of the estimated sub-continental ancestry components for individuals with >5% total Sephardic/East/South Mediterranean ancestry. In this panel colors refer to countries as for the colored country labels shown in (**a**). Following standard convention for boxplots, the center line denotes the median, the box boundaries represent the first and the third quartiles, and the whiskers range to 1.5 times the inter-quartile range on either side. Outlying points are plotted individually

a smaller Aymara sub-component (seen mostly in Southern Peruvians). In Chileans the predominant Native sub-component is most closely related to the Mapuche from Southern South America, while smaller components, related to those observed in Peruvians, are observed in Northern Chileans. In Colombians Native ancestry is most similar to Chibchan-Paezan Natives from Colombia and lower Central America, particularly in Northwestern Colombians. Other components are most closely related to the Central American Maya and, in Southern Colombians, to the Peruvian Andean Piedmont component. The overlap in Native Ancestry between Peru and neighboring Chile (to the south) and Colombia (to the north) is consistent with the high population density of the Central Andes in pre-Columbian

America, possibly associated with major cultural developments in the region (at its peak the Inca Empire extended from southern Colombia to northern Chile[29]). Finally, Andean-Piedmont ancestry from North-eastern Peru represents the major Native American contribution in the Brazilian sample (Fig. 2; Supplementary Fig. 4). Considering the low Native American ancestry in this sample compared to the other countries sampled (Fig. 1b; most Brazilians examined originate from an area of high recent European immigration[19]) and the lack of better surrogates for the Native American ancestors of current-day Brazilians (Fig. 1a), this affinity suggests a common ancestral origin between the ancestors of these Brazilians and other populations from the Amazon basin. Our results provide a high-resolution picture of

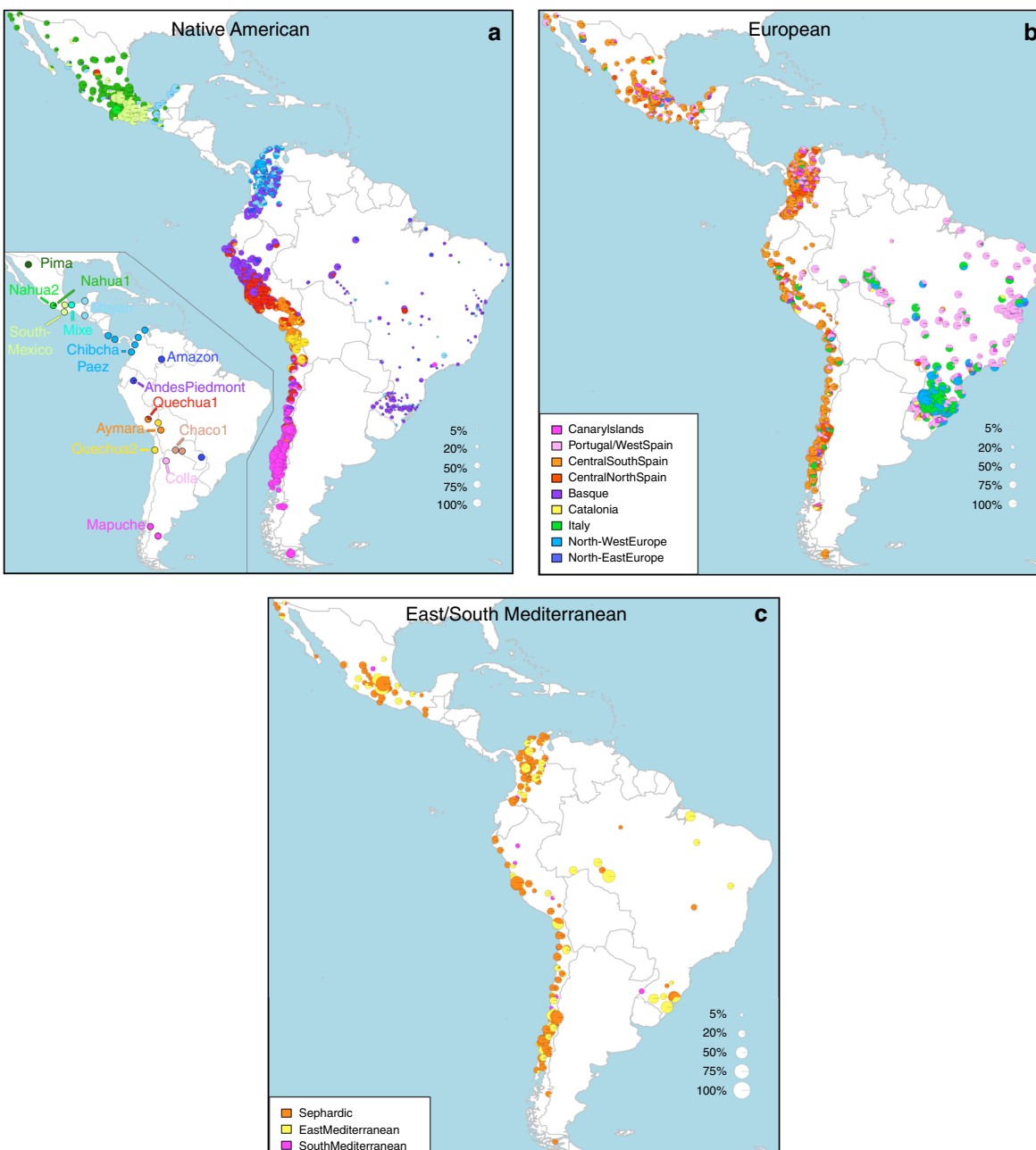

**Fig. 2** Geographic variation of ancestry sub-components in Latin American individuals. **a** Native American, **b** European, and **c** East/South Mediterranean. Each pie represents an individual, with pie location corresponding to birthplace. Since many individuals share birthplace, jittering has been performed based on pie size and how crowded an area is. Pie size is proportional to total ancestry from all sources depicted in that specific figure, and only individuals with >5% of such total ancestry are shown. Coloring of pies represents the proportion of each sub-continental component estimated for each individual (color-coded as in Fig. 1; Chaco2 does not contribute >5% to any individual and was excluded). Pies in (**c**) have been enlarged to facilitate visualization

Native variation across the Americas, emphasizing the genetic continuity between pre-Columbian groups and the Native component of present-day admixed populations across the region.

**Patterns of European ancestry in the CANDELA dataset.** Importantly, SOURCEFIND distinguishes between closely-related ancestry components from the Iberian Peninsula, as well as from the East and South Mediterranean (including individuals self-identified as Sephardic; i.e. Iberian Jews; Supplementary Note 2). The distribution of European ancestry in the CANDELA sample shows a sharp differentiation between Brazil and the

Spanish American countries (Fig. 1c). In Brazil the predominant European sub-component matches mostly the Portugal/West-Spain reference group while in Mexico, Colombia, Peru and Chile it is mostly Central/South-Spanish ancestry that is inferred (Figs. 1c and 2b). This differentiation closely matches colonial history. The European settlement of what is now Latin America involved two main areas of colonial expansion, as agreed in the Tordesillas treaty of 1494. This treaty established that territories west of a meridian somewhat east of the Amazon river-mouth were ascribed to Spain, while territories east of this meridian were attributed to Portugal[3,30]. Portuguese migration thus concentrated in Eastern South America, gradually expanding beyond

the Tordesillas meridian until achieving Brazil's current political borders, which at independence, remained a single political entity. Conversely, Spanish immigrants settled mainly in territories of Central America and Western South America, which at independence fragmented into separate countries[3]. The relatively small contribution inferred here for the Basque and Catalan agrees with historical information documenting that Spanish migrants to the Americas originated mainly in Southern and Central Spain[31]. In addition to Portugal/West-Spain ancestry the Brazilian sample also shows substantial genetic components most closely related to the Italian and German reference groups, and these concentrate in the South of the country (Fig. 2b). This pattern is consistent with the documented migration to Southern Brazil of large numbers of Germans and Italians starting in the late 19th century[30].

**Dating admixture from different sources.** To assess the timeframe of admixture between the ancestry components described above we used the program GLOBETROTTER[27]. Since admixture proportions in Latin Americans vary greatly, we analyzed each individual separately; simulations confirmed the accuracy of GLOBETROTTER in this setting (Supplementary Note 2). Inferred dates for events involving an Iberian source (the initial settlers arriving from Europe and allegedly the first to admix with the Natives) had a median of ten generations (IQR = 7–13), consistent with other estimates for admixture in Latin America[6,10,15]. Noticeably, individuals with more recent inferred dates of admixture have greater Native ancestry (Fig. 3a, Supplementary Table 4), with simulations suggesting this is consistent with continuing admixture between admixed Latin Americans and unadmixed Natives (Supplementary Note 2), possibly as a result of the decline in Iberian immigration after the mid-17th century, concomitant with the demographic recovery of neighboring Native American populations[1,32]. Compared to inferred dates related to Iberian admixture, admixture events involving non-Iberian European sources (Northwest Europe, Italy) have a significant skew towards more recent dates (Fig. 3b; Wilcoxon rank-sum test one-sided $p$-value = $3 \times 10^{-8}$), consistent with the relatively recent arrival of Germans and Italians[30].

**East/South Mediterranean ancestry in the CANDELA dataset.** SOURCEFIND finds that Sephardic/East/South Mediterranean ancestry is detectable in each country's samples: Brazil (1%), Chile (4%), Colombia (3%), Mexico (3%) and Peru (2%). Altogether, ~23% of the CANDELA individuals show >5% of such ancestry (an average of 12.2%) (Fig. 1d) and in these individuals SOURCEFIND infers this ancestry to be mostly Sephardic (7.3%), with smaller non-Sephardic East Mediterranean (3.9%) and non-Sephardic South Mediterranean (1%) contributions. Individuals with Sephardic/East/South Mediterranean ancestry were detected across Latin America (Fig. 2c). It is possible that outliers with particularly high values of Sephardic/East/South Mediterranean ancestry are descendants from recent non-European immigrants. For 19 of 42 individuals with >25% Sephardic/East/South Mediterranean ancestry, genealogical information (up to grandparents) identified ancestors born in the Eastern Mediterranean (thus validating the SOURCEFIND inference). However, no recent immigration was documented for other individuals, including all Colombians with >5% Sephardic ancestry (despite these Colombians showing the highest estimated Sephardic ancestry across countries; ~10% on average, Fig. 1d). Furthermore, GLOBE-TROTTER estimates for the time since East/South Mediterranean admixture were not significantly different from those involving Iberian sources (Fig. 3c; Wilcoxon rank-sum test one-sided $p$-value > 0.1), consistent with most of this ancestry component being contributed simultaneously with the initial colonial immigrants. Jewish communities existed in Iberia (Sepharad) since roman times and much of the peninsula was ruled by Arabs and

Berbers for most of the Middle Ages, by the end of which large Sephardic communities had developed[33]. Genetic studies have detected South and East Mediterranean ancestry in the current Spanish population, as well European admixture in the Sephardim[34–36]. The estimates of South/East Mediterranean ancestry in Latin Americans obtained here represent values over and above those present in the Iberian individuals we examined, suggesting colonial migration to Latin America involved people with relatively higher levels of South/East Mediterranean ancestry. Columbus' arrival to the New World in the late 15th century coincided with the expulsion and forced conversion of Spanish Jews, with similar measures subsequently affecting Spanish Muslims. Although Christian converts were legally forbidden from migrating to the colonies, historical records (often from the Inquisition) document that some individuals made the journey[33]. Since this migration was mostly a clandestine process, its magnitude has been difficult to assess. Genetic studies have occasionally provided evidence that certain Latin American populations could have some Converso ancestry and this is at times supported by some historical evidence[3,37,38]. Our findings indicate that the signature of a colonial migration to Latin America of people with relatively high South/East Mediterranean ancestry is much more prevalent than suggested by these special cases, or by historical records.

**Sub-Saharan African ancestry in the CANDELA dataset.** It has been estimated that Brazil received about 4.2 million African slaves (about half of those brought to the Americas) while Spanish America altogether received about 1.5 million[3]. However, the average Sub-Saharan ancestry in the full CANDELA sample is relatively low (<4%), probably reflecting the fact that regions which historically received large numbers of slaves are underrepresented in this sample (particularly for Brazil, which was sampled mainly in the South of the country)[19]. Altogether, ~22% of the individuals studied show more than 5% sub-Saharan African ancestry. SOURCEFIND infers a marked predominance of the West African sub-component, particularly in the Spanish American countries (Supplementary Figures 5 and 6), consistent with previous genetic analyses, and with historical information[1,39]. The distribution of dates involving Sub-Saharan African admixture mostly overlaps with that for Iberian admixture, although a substantial proportion of recent dates were also inferred (Fig. 3d), possibly reflecting continuing African admixture in the regions sampled.

**East Asian ancestry in the CANDELA dataset.** Other than the major Native American, European/Mediterranean and sub-Saharan African ancestry components, historical information indicates some East Asian migration to Latin America, particularly after independence in the 19th century[30]. SOURCEFIND estimates East Asian ancestry in the CANDELA sample to be, on average, very low (<1%) in Brazil, Chile, Colombia, Mexico, and slightly higher in Peru (1.4%). In individuals with >5% East Asian ancestry, this component is inferred to be most closely related to the Chinese and to a lesser extent the Japanese, except in Brazil where the opposite is found (Supplementary Fig. 7). These results match historical records documenting the arrival of Chinese laborers to Peru since the middle 19th century[40] and Japanese laborers to Brazil since the early 20th century[41]. Reflecting the relatively recent nature of these events, GLOBETROTTER estimated dates for admixture involving an East Asian source were significantly more recent than those involving Iberian sources (median = 3, IQR 2–5 generations ago, Wilcoxon rank-sum test one-sided $p$-value < $1 \times 10^{-15}$; Fig. 3e).

**Sub-continental ancestry and physical appearance.** Individuals in the CANDELA sample have been characterized for a range of physical appearance features, including aspects of anthropometry,

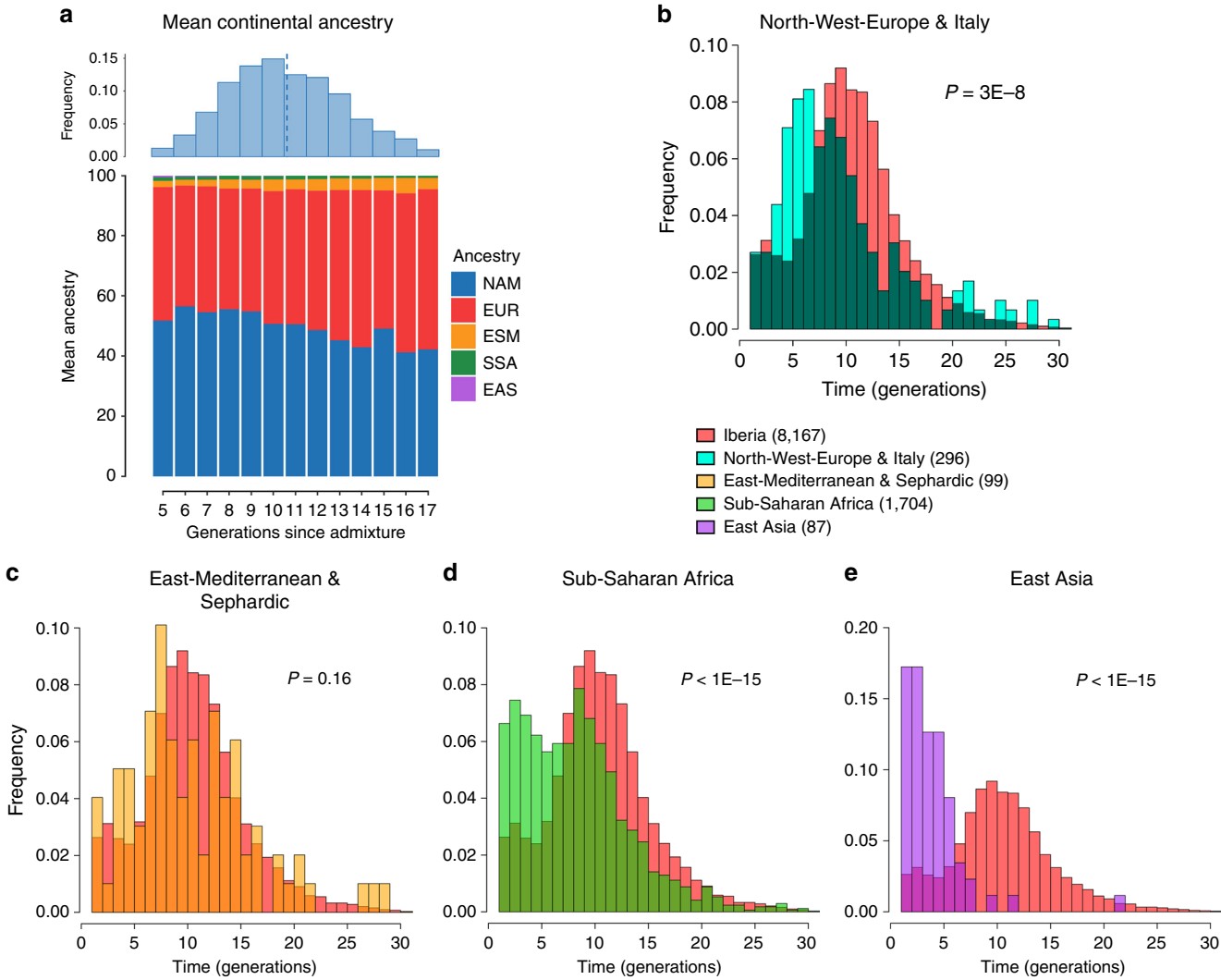

**Fig. 3** Times since admixture estimated using GLOBETROTTER. **a** Top: frequency distribution of admixture times for individuals in which a single admixture event between Native and European sources was inferred (dashed line indicates the mean). Bottom: mean continental ancestry (%) as a function of time since admixture among these individuals. Only time bins including >20 individuals are shown. (NAM Native American, EUR European, ESM East/South Mediterranean, SSA Sub-Saharan African, EAS East Asian). **b–e** show contrasts of the distribution of admixture times involving Iberian versus other sources: **b** North-West Europe and Italy, **c** East Mediterranean and Sephardic, **d** Sub-Saharan Africa and **e** East Asia. *p*-values for comparing the mean date of Iberian versus each other ancestry source are from a one-sided Mann–Whitney *U* test, and numbers of inferred admixture events are given in parenthesis

face and ear morphology, facial and scalp hair, and pigmentation (of hair, skin and eyes) (Supplementary Note 4). We evaluated the impact of sub-continental genetic ancestry on these features using linear regression. To maximize power and reduce collinearity, we focused on contrasts involving the most frequent and differentiated sub-continental ancestry components (see Methods, Fig. 1). SOURCEFIND results allowed the analysis of two contrasts. The first involved North-West Europe versus Portugal/West-Spain ancestry in the Brazilian sample. We observed a highly significant effect of this contrast on pigmentation traits (Fig. 4a–c). This observation validates our approach, as it is consistent with the latitudinal gradient in pigmentation observed within Europe, and the corresponding differentiation in allele frequencies at pigmentation genes between Northern and Southern Europeans[42]. The second contrast examined involved a Central Andean component (obtained by merging the closely-related Quechua1, Quechua2, Colla and Aymara components) versus the relatively differentiated Mapuche component (Fig. 1). This contrast is significantly associated in the CANDELA sample, with variation in facial features, particularly nose shape (Fig. 4a,

b, d). Validation analyses limited to Peru and Chile or only to Chile, using the ancestry components inferred by SOURCEFIND as well as related components obtained with ADMIXTURE or PCA (Supplementary Figures 8 and 9, Supplementary Note 1), produced similar results (Fig. 4e, Supplementary Note 5).

It is noticeable that regional Native American ancestry impacts on nose shape. The Mapuche component is strongly associated with a less protruded nose (*p*-value $<2 \times 10^{-5}$) and broader nose tip angle (*p*-value $< 10^{-7}$). This is consistent with physical anthropology studies indicating that the Mapuche have a flatter, wider nose than Central Andean populations[43]. In a recent GWAS for facial features in the CANDELA sample, most loci identified impacted on nose shape[21]. For each of the six index SNPs significantly associated with facial features in that GWAS, allele frequencies at haplotypes inferred to be of Central Andean ancestry were significantly different from allele frequencies at haplotypes inferred to be of Mapuche ancestry (Supplementary Table 5). Furthermore, for each of these six SNPs, the frequency of the allele associated with an increase of the phenotypic trait

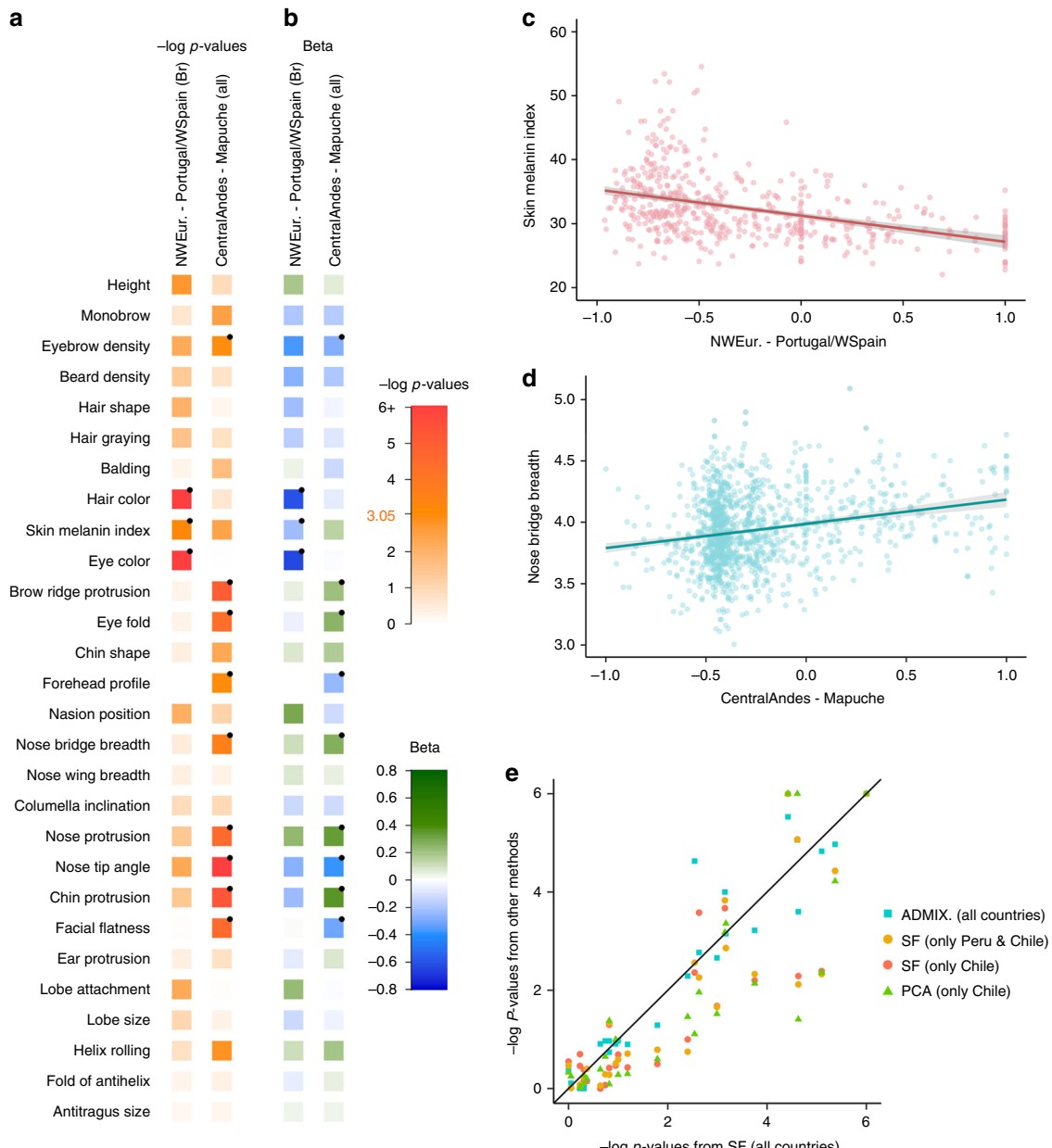

**Fig. 4** Effect of sub-continental genetic ancestry on physical appearance. **a** Regression –log *p*-values for 28 traits (Supplementary Note 4) against the contrast between two sub-continental ancestry components estimated by SOURCEFIND. The left column shows results for the Portugal/West-Spain versus North-West Europe contrast in the Brazilian sample (Br). The right column presents the contrast between Central Andes versus Mapuche ancestry in the full CANDELA sample. **b** Regression coefficients (Betas) in units of SD for the contrasts in (**a**). In **a**, **b** color intensity reflects variation in -Log-*p* values or beta coefficients, as indicated on the scale. Bonferroni-corrected significant values are highlighted with a dot (–log *p*-value threshold of 3.05 for alpha = 0.05). **c**, **d** Display scatterplots and regression lines (with 95% confidence intervals) for two traits showing significant association with variation in sub-continental ancestry: skin melanin index in Brazilians (**c**) and nose bridge breadth in Chileans and Peruvians (**d**; *Y*-axis is in Procrustes units). **e** Scatterplot of -log *p*-values from follow-up analyses of the regression of physical traits on the Central Andes versus Mapuche ancestry contrast. The X-axis refers to -log *p*-values from the primary analyses (using SOURCEFIND (SF) estimates and data for all individuals, as shown in the second column of (**a**)). The Y-axis refers to -log-*p* values from four other regression analyses: using SOURCEFIND (SF) estimates restricted to Peruvian and Chilean individuals, or only to Chileans; using related ancestry components defined by: ADMIXTURE (ADMIX., at *K* = 7) in all the CANDELA data, or by PCA (PC 7), in an analysis limited to Chileans (Supplementary Note 5, Supplementary Figures 8 and 9). Sample sizes: all data *N* = 5,794, Peruvians and Chileans *N* = 2,594, Chileans *N* = 1,542

was higher in the Native component associated with an increase of that trait. The nasal cavity is an important regulator of inhaled air temperature and humidity, and evolutionary studies suggest that nose shape has been influenced by adaptation to cold/dry versus hot/humid environments[44]. Consistent with selection effects at these SNPs, allele frequencies at the set of six

GWAS index SNPs jointly were more differentiated between Central Andean and Mapuche than was the case in randomly selected sets of six genome-wide SNPs that matched each index SNP for the number of inferred Native ancestry haplotypes and minor-allele frequency in either the inferred Central Andean (*p*-value < 0.02) or inferred Mapuche (*p*-value < 0.01) haplotypes

(Supplementary Figures 10 and 11, Supplementary Table 6). Since variation in altitude correlates with air temperature and humidity, it will be interesting to explore further whether the association of Central Andean ancestry with nose shape and the differences in allele frequencies in loci associated with facial features between Central Andeans and the Mapuche relate to altitude adaptation during Native American evolution.

## Discussion

By leveraging information from shared haplotypes, here we infer the timings and proportions of ancestry contributions to Latin Americans since the colonial era. While previous work has suggested GLOBETROTTER's inferred dates are robust to using different surrogates to the true ancestry sources[27], inferred proportions of ancestry inevitably depend on which surrogate groups are used. In general our SOURCEFIND inference suggests that the reference populations included in this study are good surrogates of the true ancestral sources, as demonstrated by the preferential matching to specific geographic regions of Iberia (Fig. 2b) and the strong correspondence between geography and ancestry matching in the Native component (Fig. 2a). A caveat to this is that some of our reference Native groups evidenced strong genetic drift and SOURCEFIND inferred negligible contributions from such groups (Supplementary Table 7). Indeed if such drift is post-Columbian, the extant Native populations may not represent well the pre-Columbian Natives that admixed with immigrant settlers. DNA from the remains of pre-Columbian Native Americans could shed light on the extent to which this might be the case.

A further complication is that some of the reference populations may have experienced admixture following the colonial period. For example, it is possible that the Iberian reference individuals examined here have less non-European (i.e. East/South Mediterranean and/or Sub-Saharan African) ancestry than individuals migrating to the Americas during the colonial period, due to more recent admixture with other Europeans. In this case SOURCEFIND may overestimate the contributions from the non-European groups. Because of this, estimates for each of the East/South /Mediterranean and African components should be interpreted as values over and above those present in the present-day Spanish/Portuguese reference individuals examined. As noted above, the similarity in inferred dates for admixture involving East/South Mediterranean versus Iberian ancestry furthermore suggests that the individuals carrying this excess East/South Mediterranean ancestry migrated to Latin America during the colonial period.

In conclusion, the results presented here exemplify how historical events have finely structured the genetic make-up of Latin Americans, and provide insights into the complicated dynamics and timescales of intermixing among different continental groups from the colonial-period up until recently. Our findings illustrate how genetic analyses can contribute to building a fuller picture of human history. This is particularly the case for poorly documented events such as the clandestine migration of recent Christian Conversos, of East/South Mediterranean ancestry, to colonial Latin America. Furthermore, our analyses show how regional genetic variation, subtly shaped by history, can impact on the genetic architecture of complex phenotypes across major geographic regions. We demonstrate how this regional genetic diversity can be analyzed in admixed individuals with ancestry from various sources; an encouraging result given the ubiquity of recent admixture in world-wide populations[27,45]. Our results underline the importance, for a fuller exploitation of genomic data, of a broader description of human genetic and phenotypic diversity than is currently available.

## Methods

**Genotype datasets**. The CANDELA dataset (http://www.ucl.ac.uk/candela) consists of genotypes from 6,852 individuals ascertained in five Latin American

countries (Brazil $N = 676$, Chile $N = 1,891$, Colombia $N = 1,713$, Mexico $N = 1,288$ and Peru $N = 1,284$) (Supplementary Fig. 1). This study sample has been described in detail in Ruiz-Linares et al.[19]. Briefly, adult individuals of both sexes were ascertained at one main recruitment site per country (Porto Alegre in Brazil, Arica in Chile, Medellín in Colombia, Mexico City in Mexico, and Lima in Peru). Informed consent was obtained from all participants, complying with all relevant ethical regulations as approved by Universidad Nacional Autónoma de México (México), Universidad de Antioquia (Colombia), Universidad Perúana Cayetano Heredia (Perú), Universidad de Tarapacá (Chile), Universidade Federal do Rio Grande do Sul (Brazil) and University College London (UK). A structured interview recorded the birthplace of volunteers and their ancestors (up to grandparents), as well as information on the language(s) spoken by them. We have previously reported genome-wide association studies based on Illumina HumanOmniExpress chip data obtained in these individuals[20–22].

To perform ancestry analyses in the CANDELA individuals we collated a reference population dataset from regions having potentially contributed to admixture in Latin America. We combined publicly available data[46–50] with data from newly genotyped samples obtained here (Fig. 1, Supplementary Table 1, Supplementary Fig. 2). Altogether we collated data for 2,359 individuals from 117 reference populations (38 Native American, 42 European, 15 East/South Mediterranean, 15 Sub-Saharan African, and 7 East Asian). Of these, 42 were newly genotyped population samples (comprising 27 Native American, seven European, and eight East/South Mediterranean), including a total of 430 individuals. These individuals were genotyped on the Illumina HumanOmniExpress chip which includes 730,525 SNPs. PLINK v1.9[51,52] was used to exclude SNPs and individuals with more than 5% missing data, markers with minor-allele frequency <1%, related individuals, and those who failed the X-chromosome sex concordance check. The same QC filters had been applied to the CANDELA dataset[20–22]. Individuals born outside the country were relocated when coming from one of the five countries included in this study or otherwise removed. Similar quality controls were applied to the public reference population datasets. In addition, unsupervised ADMIXTURE[25] analyses of reference population samples were used to identify and exclude Sub-Saharan Africans, East Asians, and Europeans with less than 95% of their own continental ancestry. In the case of Native Americans, all individuals were initially retained (regardless of admixture levels), but reference individuals with less than 95% Native American ancestry were only used for haplotype phase inference. In the case of East/South Mediterranean individuals, ADMIXTURE consistently inferred Sub-Saharan African ancestry. The estimated Sub-Saharan African ancestry proportions were found to be quite homogeneous across individuals, possibly indicating relatively old shared ancestry. Based on this assumption, we excluded only four individuals with admixture proportions deviating markedly from this observation in a manner suggestive of recent admixture (three Moroccans with Sub-Saharan African ancestry >40% and one Libyan with Sub-Saharan African ancestry of 79%; both of these populations have an estimated average Sub-Saharan African ancestry of ~20% with a standard deviation of $+/-3\%$).

After QC, the merged CANDELA plus reference population dataset comprised genotypes for 546,780 autosomal SNPs in 8,647 individuals (including 6,589 Latin Americans and 2,058 individuals from the reference population samples). A global overview of our analysis strategy is provided in Supplementary Figure 12.

**Phasing of genotype data**. Phasing of the merged dataset was performed with SHAPEIT2[53] using default parameters. Genetic distances used were obtained from the HapMap Phase II genetic map build GRCh37[54]. Missing genotypes for any SNP (<5% after the QC) were imputed during the phasing process.

**Inference of haplotype similarity patterns**. CHROMOPAINTER[23] was used to infer haplotype similarity (informally, chromosome painting) and estimate the proportion of DNA shared between donor and recipient individuals, thus generating a coancestry matrix relating individuals. CHROMOPAINTER was setup so that the donors were exclusively reference population individuals, while recipients included CANDELA individuals as well as reference population individuals. Consideration of reference population individuals as recipients as well as donors enabled the analysis of genetic structure in the reference population samples (using fineSTRUCTURE as described below), while the analysis of coancestry between CANDELA recipients and reference population donors enabled the inference of ancestry profiles in the CANDELA individuals from donor clusters (using SOURCEFIND as described below).

The recombination scaling constant $N_e$ and the mutation parameter $\theta$ used by CHROMOPAINTER were jointly estimated for every individual in a subset of chromosomes (1, 6, 13, and 22) with 10 Expectation-Maximization steps, starting from default values defined by the software. The average $N_e$ and $\theta$ values across chromosomes (weighted by chromosome size) were then used for subsequent CHROMOPAINTER runs on all autosomes ($N_e = 290.83$ and $\theta = 0.00038$). Genetic distances from the HapMap Phase II genetic map build GRCh37 were used in the CHROMOPAINTER runs. CANDELA individuals with >99% European ancestry (52 Brazilians, of which 37 reported German and 15 Italian ancestors) or with >95% Native American ancestry (1 Colombian, 22 Mexicans, 65 Chileans and 17 Peruvians) were included amongst the donors as they may harbor ancestry components not present in our reference dataset. In Supplementary Note 6 we

show that our conclusions about ancestry are similar if these CANDELA individuals are excluded from the reference dataset. In total, 157 CANDELA individuals and 1,942 reference individuals were added to the panel of donors, for a total of 2,099 samples. The remaining 116 individuals from the initial reference dataset were excluded. Of these 80 were Native Americans with less than 95% Native ancestry, and 36 were Native Americans excluded after the haplotype-based clustering analyses performed to select the reference panel for the ancestry inference, as explained in the next section.

**Definition of clusters of reference population individuals**. To evaluate genetic structure in the reference populations independent of sample labels we used fineSTRUCTURE[23], a program that defines homogeneous clusters of individuals based on the coancestry matrix produced by CHROMOPAINTER. We performed additional analyses on the clusters defined by fineSTRUCTURE in order to select a final subset of donor clusters to be used as surrogates for the unknown populations that historically contributed ancestry to Latin Americans (we refer to this subset as surrogate clusters or surrogates). These additional analyses aimed to: (i) reduce the number of clusters potentially representing sources of ancestry, (ii) avoid problems related to collinearity between the different sources when estimating ancestry, and (iii) facilitate interpretation of results. The end result was that 56 surrogate clusters were defined and subsequently used for sub-continental ancestry estimation in the CANDELA samples. The fineSTRUCTURE analyses and their subsequent refinement are described below and a diagram summarizing the overall strategy is shown in Supplementary Fig. 13. Supplementary Table 2 summarises the specific criteria defining each donor cluster.

First, the reference population individuals were clustered using fineSTRUCTURE. A likelihood adjustment factor (c) is initially calculated in order to account for the inaccurate assumption that the amount of DNA matching among individuals is independent. Using default CHROMOPAINTER settings to infer the adjustment factor, this was estimated as $c = 0.236$. Two MCMC runs were performed using 1,000,000 iterations (sampling every 10,000, after 1,000,000 burn-in iterations). The first run was used as the baseline to define the clusters, while the second one was used to assess the variability of the cluster assignments between runs (described below). Following Leslie et al.[24], for each run the sample with maximum posterior probability was selected and an additional 100,000 hill-climbing moves were then performed to search for merges or splits that further improve the overall model likelihood[23]. After this procedure, fineSTRUCTURE classified individuals in 129 clusters.

Using the 129 clusters defined by fineSTRUCTURE, we performed a preliminary estimation of sub-continental ancestry in the CANDELA samples using a modification of the NNLS approach implemented by Hellenthal et al.[27] and Leslie et al.[24]. We performed this analysis (and the further refinements described below) using NNLS (instead of SOURCEFIND; described below) so as to identify the surrogate clusters detected in a computationally efficient manner. Based on this preliminary analysis, we outlined criteria to define certain surrogate clusters (as described below). Note that even though only surrogate clusters are used for estimating ancestry in CANDELA samples, our modelling requires use of the full coancestry matrix, implying that individuals from non-surrogate clusters are still included as donors in order to define the haplotype similarity profiles for both surrogates and admixed samples.

We checked the consistency of the assignments of individuals to clusters across samples in the 1,000,000 iterations of the two fineSTRUCTURE runs. We excluded individuals that were assigned to different clusters more than 10% of the time. This included five clusters consisting exclusively of individuals that were inconsistently assigned across samples. We also excluded: (i) 12 clusters consisting of only one sample, (ii) ten small clusters with low ancestry contributions to CANDELA samples (furthermore, other individuals with the same population sample labels formed larger independent clusters), and (iii) 17 clusters that did not contribute to the CANDELA samples. Altogether, this led to 44 clusters being excluded as surrogates. Since these 44 clusters had no clearly defined structure, the individuals they included were reclassified, based on population sample labels, into 54 non-surrogate donor clusters. Furthermore, we merged 16 clusters with other clusters containing mainly individuals from the same reference population sample. NNLS usually randomly assigned ancestry to Latin Americans from these 16 clusters and the related, larger, clusters (probably reflecting their genetic similarity, e.g. two clusters made of Nigeria.1 samples; see Supplementary Table 2 for details). Generally these 16 clusters also showed low Total Variation Distance (TVD) (e.g. as used in Leslie et al.[24]) and Tree distance values from their larger, related clusters.

In sum, these refinements resulted in a total of 69 surrogate and 54 non-surrogate donor clusters being defined.

We next used the modified NNLS regression approach to check if certain of the 69 surrogate donor clusters could introduce collinearity issues in subsequent analyses (due to high relatedness) or had complex ancestry profiles complicating the interpretation of results. For this, the proportions of DNA that each individual from the surrogate clusters matches to each donor individual (estimated by CHROMOPAINTER) were summed across the 123 non-surrogate donor clusters defined above (resulting in a 123-variable vector that we call a copying vector). We obtained the average copying vectors (across individuals) for each of the 69 surrogate clusters. We then performed a NNLS regression with the average copying vector of a surrogate cluster as the response and the average copying vectors the other 68 clusters as predictors. These analyses detected:

(i) seven (7) clusters that contributed substantial ancestry to several other surrogate groups (e.g. a Sardinia cluster contributes ~15% ancestry to the Portugal/WestSpain, Catalonia, and Italy clusters) or showed ancestry from more than one continent (e.g. a Turkey cluster was inferred to have >5% ancestry from East Asia and 5% from Europe). These seven clusters were excluded from the surrogates and subsequently considered as non-surrogate donor clusters.

(ii) Six (6) Native American clusters (including Uros, Kogi, Karitiana, Surui, Ticuna and Mixe individuals; Supplementary Table 2) that showed evidence of strong genetic drift (high haplotype similarity within a cluster and their copying profile could not be explained by mixtures of other donors) and that contributed little ancestry to the CANDELA samples. These six clusters were removed from both surrogates and donors, in an attempt to mitigate the effect of genetic drift in their haplotype similarity profiles and use these clusters exclusively as surrogates (without being donors), but this procedure had no effect on estimated ancestry proportions.

The end result was that the initial set of 69 surrogate clusters was reduced to 56, which was the final set of surrogates used for the ancestry analyses of the CANDELA individuals, and the total set of donors clusters (surrogates and non-surrogates) reduced to 117. The 56 surrogate clusters include a total of 1,444 reference population individuals. Supplementary Table 3 details the individual makeup of these 56 clusters in terms of the reference population labels. Supplementary Figure 3 shows a phylogenetic tree relating these clusters and detailing the 35 surrogate groups that were defined (based on cluster relatedness) to facilitate the display of ancestry profiles of the CANDELA individuals (Fig. 1).

The maps in Figs. 1 and 2 and Supplementary Information were drawn with the statistical software R[55] using packages rworldmap, maptools, and plotrix.

**A new haplotype-based estimation of ancestry**. The 56 surrogate clusters defined above were used for inferring the ancestral population contributions to admixture in Latin America. We generated copying vectors for each CANDELA individual and for each individual included in the 56 surrogate clusters by summing the proportion of DNA that every individual matched to individuals from the 117 donor clusters defined in the previous section. To cope with differences in surrogate cluster size and improve resolution, we modelled the copying vector of each CANDELA individual as a weighted mixture of the copying vectors from the surrogates[24,27]. To do so, we introduce a model-based approach we term SOURCEFIND (see code availability), which outperformed the NNLS approach taken in Leslie et al.[24] in simulations related to this study (see Supplementary Note 2). Below we describe the SOURCEFIND algorithm.

Let $l^r \equiv \{l_1^r, \dots, l_D^r\}$ be the copying vector describing the total genome length (in cM) that a recipient individual (or group) $r$ copies from each of the $d \in [1,\dots, D] = 117$ donor clusters as inferred by CHROMOPAINTER (Note that copying vectors can also be averaged across recipients to perform the analysis in groups). Here for any $r$, $\sum_{d=1}^{D} l_d^r = C$, where $C$ is equal to the total genome length of DNA (in cM), times two because we sum matching across a recipient's two haploid genomes, and we further define $f_d^r \equiv \frac{l_d^r}{C}$. Henceforth we let $r$ denote a CANDELA individual, and s denote a surrogate cluster. In the latter case, $l_d^s$ represents an average across all individuals from that surrogate cluster.

We assume that:

$$\Pr(l^r | l^1, \dots, l^S, C, \beta^r) = \text{Multinomial}\left(C; \sum_{s=1}^{S} [\beta_s^r f_1^s], \dots, \sum_{s=1}^{S} [\beta_s^r f_D^s]\right) \quad (1)$$

where $\beta^r \equiv \{\beta_1^r, \dots, \beta_S^r\}$ are the mixture coefficients we aim to infer and every $s \in [1,\dots,S] = 56$ represents a surrogate cluster used to describe the ancestry of group $r$. In practice, often all the donor clusters are used as surrogates, so that $S = D$. However, in our case the surrogates are a subset of the donors so that $S < D$.

We take a Bayesian approach to inferring $\beta^r$, further assuming the following:

$$\Pr(\beta^r | \lambda) = \text{Dirichlet}(\lambda_1, \dots, \lambda_S), \quad (2)$$

$$\Pr(\lambda_s) = \text{Uniform}(0, 10), \quad (3)$$

where $\lambda = \{\lambda_1,\dots,\lambda_S\}$. For each recipient $r$, we wish to sample the mixing coefficients $\{\beta_1^r, \dots, \beta_S^r\}$ based on their posterior probabilities conditional on $l \equiv \{l^r, l^1,\dots,l^S\}$. We do so using the following Markov Chain Monte Carlo (MCMC) technique. We start with an initial value of $\lambda(0) = 0.5$ and sample our initial values of $\beta^r(0) \equiv \{\beta_1^r(0), \dots, \beta_S^r(0)\}$ from the prior distribution Dirichlet $(\lambda(0),\dots,\lambda(0))$. Then we perform the following for $m \in [1,\dots,M]$, where $M$ is the total number of MCMC iterations:

Update $\beta^r(m) \equiv \{\beta_1^r(m), \dots, \beta_S^r(m)\}$ using a Metropolis-Hastings (M-H) step:

i. Randomly sample $Y \sim Unif (0,0.1)$.
ii. Randomly sample a surrogate $s_x$ and set $\beta_{s_x}^r (m) = \beta_{s_x}^r (m-1) + Y/5$. For numerical stability, if $\beta_{s_x}^r(m) > 1 - 1e^{-7}$, set $\beta_{s_x}^r(m) = 1 - 1e^{-7}$.

Repeat this for four additional randomly sampled (with replacement) surrogates $s_x$.

iii.   Randomly sample a surrogate $s_x$ and set $\beta^r_{s_x}(m) = \beta^r_{s_x}(m-1) - Y/5$. For numerical stability, if $\beta^r_{s_x}(m) < 1 - 1e^{-7}$, set $\beta^r_{s_x}(m) = 1e^{-7}$.
Repeat this for four additional randomly sampled (with replacement) surrogates $s_x$.

iv.   For all other surrogates $s \in [1,\dots,S]$, excluding the randomly sampled set above, set $\beta^r_s(m) = \beta^r_s(m-1)$.

v.   Re-scale $\sum_{s=1}^{S} \beta^r_s(m) = 1.0$.

vi.   Accept $\beta^r(m)$ with probability $\min(\alpha, 1.0)$, where:

$$\alpha = \frac{Pr(l^r | l^1, \dots, l^S, C, \beta^r(m)) Pr(\beta^r(m) | \lambda(m-1))}{Pr(l^r | l^1, \dots, l^S, C, \beta^r(m-1)) Pr(\beta^r(m-1) | \lambda(m-1))}.$$

Update each $\lambda_s(m)$ for $s = 1,\dots,S$ using a M-H step:

i.   Propose a new $\lambda_s(m)$ from a Normal $(\lambda_s(m-1), sd = 0.2)$.
ii.   Automatically reject if $\lambda_s(m) \notin [0,10]$.
iii.   Otherwise accept $\lambda_s(m)$ with probability $\min(\alpha, 1.0)$), where:

$$\alpha = \frac{Pr(\beta^r(m) | \lambda(m))}{Pr(\beta^r(m) | \lambda(m-1))}.$$

For large $M$, this algorithm is guaranteed to converge to the true posterior distribution of the $\beta^r$s (e.g. Gamerman[56]). In practice, we used $M = 200,000$, sampling every 1000 iterations. Also, for each recipient individual $r$, we combined results across 50 independent runs of the above procedure, extracting the estimates with the highest posterior probability in each run and then taking a weighted (by posterior probability) average of these 50 estimates. We refer to the final estimates of $\{\beta^r_1, \dots, \beta^r_S\}$, weighted by posterior values, as our inferred proportions of ancestry for individual $r$ conditional on this set of $S$ surrogates. This approach differs from the mixture model procedure applied in previous studies[13,24,27,57,58] in that it assumes that $l^r$ is multinomial distributed and solves for $\beta^r$ using a Bayesian approach rather than a NNLS optimization. The model is similar to the one described by Broushaki et al.[59], but introduces new improvements in the way that $\lambda$ is estimated and in the MCMC proposal procedure.

**Estimation of the number of generations since admixture.** The times and sources of major admixture events were inferred using the program GLOBE-TROTTER[27]. GLOBETROTTER tests for evidence of one or more pulses of admixture between two or more ancestral groups, and dates these admixture events and infers the genetic make-up of the admixing groups involved. Due to the recent nature of intermixing in the Americas, admixture times and proportions may vary substantially across CANDELA individuals. Therefore we tested each individual separately, restricting this analysis to the 6,352 individuals inferred by SOURCE-FIND to have ancestry from more than one surrogate cluster.

For each haploid genome of each individual, we used ten random samples of genome-wide local matching to donor clusters per haploid as provided by the CHROMOPAINTER analysis described above. For each CANDELA individual, we ran GLOBETROTTER including as surrogates only the subset of ≤56 clusters that contributed >1% to that individual, as inferred by SOURCEFIND. For each CANDELA individual, GLOBETROTTER categorized admixture inference into one of three types: (i) one date of admixture involving two sources, (ii) one date of admixture involving more than two sources (suggestive of a admixture among multiple genetically different groups within a short time span), and (iii) multiple dates of admixture between two or more sources (not necessarily the same two), suggesting a more complicated history but which GLOBETROTTER attempts to describe as two major pulses of admixture.

Altogether, for 55.4% of the CANDELA individuals (3,519/6,352) GLOBETROTTER inferred a single admixture event between two source groups, while in 44.6% of the CANDELA individuals (2,833/6,352) a more complex admixture was inferred. This could consist of more than two groups admixing (Supplementary Fig. 14) and/or multiple dates of admixture (Fig. 3b, Supplementary Table 8). For simplicity, the inferred admixture history for these complex admixture events was described as two distinct events, with each event characterised as having two admixing groups and a single date of admixture. In total GLOBETROTTER inferred 9,185 such admixture events (Supplementary Table 8). For simplicity, we represent the two admixing sources using GLOBETROTTER's best-guess results, which describes each admixing source by the single (included) surrogate group out of 56 that is inferred to be most genetically similar to that (unknown) admixing source group.

To convert the time estimates obtained by GLOBETROTTER (in generations) into years, we used the formula $y = 1990 - 28 \times (g + 1)$, where y is the year of admixture, 1990 is the mean birth year in CANDELA individuals, g the estimated admixture time (in generations), and taking 28 years as the generation time[60].

**Differences in inferred admixture dates by source groups.** In Fig. 3, we plot histograms of inferred dates for each of the major geographic labels Iberia, NorthWestEurope & Italy, East Mediterranean & Sephardic, Sub-Saharan African (SSA) and East Asia. These plots contain the inferred dates for all admixture events (out of 9,185) that involved an inferred source group categorized under that major geographic label, with:
Iberia: CanaryIslands, Portugal/WestSpain, CentralSouthSpain, CentralNorthSpain, Basque and Catalonia.
NorthWestEurope & Italy: Italy1 and NorthWestEurope1.
East Mediterranean & Sephardic: Sephardic1, EastMediterranean1 and EastMediterranean2.
Sub Saharan Africa: WestAfrica1, WestAfrica2, WestAfrica3, EastAfrica1, EastAfrica2, Namibia, and SouthAfrica.
East Asia: Japan, ChinaHan, China/Vietnam1, and China/Vietnam2.

We used wilcox.test in R[55] to perform a one-sided Wilcoxon rank-sum test (also known as a Mann–Whitney $U$ test) to test the alternative hypothesis that the distribution of admixture dates for each geographic label X = {East Asia, NorthWestEurope & Italy, East Mediterranean & Sephardic, SSA} is skewed towards more recent dates relative to the Iberia geographic label, versus the null hypothesis that distributions are the same. Though they may represent genuine admixture events, for these tests and the histograms of Fig. 3 we removed events with an inferred date of 1. This was done both to avoid such dates dominating inference due to their high frequency (8% of all events in Iberia have inferred dates of 1, with East Asia = 21%, NorthWestEurope & Italy = 6%, East Mediterranean & Sephardic = 10%, SSA = 13%) and because such events have been interpreted as evidence of no admixture in past applications of GLOBETROTTER[27]. For the Wilcoxon rank-sum test, we further excluded individuals with ≤5% ancestry from X and individuals with date ≥ 30 generations to avoid admixture events that occurred prior to colonial-era migrations. In addition, this analysis assumes each inferred event is an independent observation, even though some individuals have two inferred events. However, we note that conclusions and trends do not change if we restrict to one inferred event per individual, e.g by excluding individuals who infer multiple dates of admixture (i.e. case (iii) described in the previous section) and only including the more strongly signaled event in individuals who infer more than two sources of admixture at the same time (i.e. case (ii) described in the previous section).

**Association of sub-continental ancestry with physical traits.** We recorded 28 physical appearance traits, by physical examination of the volunteers and/or by examining facial photographs. These traits have been described in detail in previous studies[19–22] and brief definitions are provided in Supplementary text 5.

To evaluate the phenotypic effect of sub-continental ancestry components defined by SOURCEFIND we used linear regression. Since these components are (negatively) correlated with other major continental ancestries, using them directly would cause confounding in the linear model. We therefore performed linear regression analysis in a manner analogous to that in Moreno-Estrada et al.[14], but including a contrast between subcontinental ancestry components. To maximize power, we defined three criteria for making these contrasts: (i) each component tested should have at least 10% frequency in a country (ii) the two sub-continental ancestry components contrasted should add up to at least half of the total continental ancestry in a country, and (iii) the components contrasted should show a relatively high genetic differentiation.

These criteria only allowed one contrast to be made based on the European components (Fig. 1): that between North-West Europe and Portugal/West-Spain in Brazil. In addition, merging the closely-related Quechua1, Quechua2, Colla and Aymara into a Central Andean component, enabled a Native American contrast based on the SOURCEFIND analysis. Similar components were defined by Principal Component (PC) 7 (Supplementary Fig. 9) and by ADMIXTURE at K = 7 (Supplementary Fig. 8), which we tested for consistency.

The basic regression model tested was:

$$\text{Phenotype} \sim \text{Age} + \text{Sex} + \text{Socioeconomic status}$$
$$+ \text{Total Sub Saharan African ancestry} + \text{Total European ancestry}$$
$$+ \text{Native component contrast,}$$

or,

$$\text{Phenotype} \sim \text{Age} + \text{Sex} + \text{Socioeconomic status} + \text{Total Sub Saharan African ancestry}$$
$$+ \text{Total Native American ancestry} + \text{European component contrast.}$$

For facial traits, BMI was included as a covariate. When doing a multi-country analysis we also used country as dummy variable. To reduce variability from other continental ancestries, we excluded individuals with high Sub Saharan African or East/South Mediterranean ancestry and individuals with >1% East Asian ancestry.

**Differences in allele frequencies of GWAS index SNPs.** To test whether allele frequencies differed between individuals with Mapuche versus Central Andean ancestry at loci previously identified as being associated with facial features[21], we

first inferred the allele frequencies at these loci in each of the Mapuche and Central Andean populations. As we have relatively few reference individuals with Mapuche and Central Andean ancestry, we inferred allele frequencies by combining these reference samples with admixed Candela individuals that were inferred to carry the appropriate Native ancestry at these loci.

To do so, we used the software RFMix[61] to infer local continental ancestry in the subset of phased CANDELA individuals described earlier. Three continental reference panels (consisting of phased haplotypes for 107 IBS (Iberian Population in Spain; 1000 Genomes Project), 101 YRI (Yoruba in Ibadan, Nigeria; 1000 Genomes Project) and 125 Native American samples) were used for this purpose. RFMix assigns local continental ancestry to each allele of each CANDELA haplotype, allowing for errors in genotyping, slight admixture in the reference samples, etc. Thus, for each allele of each haplotype, it produces two files of relevance—the local ancestry at that site, and the putative allele at that site (after fixing any such errors).

Using SOURCEFIND sub-continental ancestry proportions, two different sets of CANDELA individuals were selected to obtain allele frequencies for Central Andes and Mapuche groups. For each set, all individuals had >10% inferred ancestry from that Native group, with <1% combined inferred ancestry from all other Native groups and <1% inferred East Asian ancestry. For all individuals in a group, for each locus, all alleles that had local Native ancestry (as inferred by RFMix) were aggregated to estimate the allele frequency for that group. Allele frequencies thus obtained for Central Andes were very similar to the allele frequencies obtained from 49 surrogate individuals of the Central Andes group who were inferred to have >99% Native ancestry ($r^2 > 0.99$; the number of surrogate individuals with >99% Native ancestry for the Mapuche group wasn't large enough for such a comparison).

Allele frequencies were thus obtained for the index SNPs (among the chip data) of all the six genomic regions identified in Adhikari et al.[21]. A two-sample t-test (assuming unequal means and variances) was used to assess whether the allele frequencies were significantly different in Central Andes vs. Mapuche individuals. The FDR (false discovery rate) procedure was used to control the Type-I error rate at 0.05 level. After the FDR procedure, all SNPs showed a significant difference in allele frequency between Central Andes & Mapuche (Supplementary Table 5). Furthermore, for each SNP, the allele with a higher frequency in Central Andes compared to Mapuche had the same direction of effect (same signs of regression coefficient beta) for that allele in the GWAS as compared to the regression coefficient (beta, Fig. 4b) between the Central Andes-Mapuche contrast and the trait.

We also assessed whether the allele frequencies at these six SNPs jointly were excessively differentiated between haplotypes inferred to be of Central Andean ancestry versus those inferred to be of Mapuche ancestry, potentially indicating selection. To do so, we randomly selected sets of six genome-wide SNPs. For each SNP in the set of six, we used the same t-test to calculate a p-value testing the null hypothesis that the Central Andean and Mapuche allele frequencies were the same, taking the average −log p-value across all six SNPs in the set. We found the proportion of 10,000 such random samples of six SNPs with average −log p-value less than or equal to that of the six GWAS index SNPs, using this proportion as an empirical p-value testing whether the six GWAS index SNPs were more differentiated than usual. In order to match power between our six GWAS hit SNPs and each random set of six SNPs, we only randomly selected from SNPs matched to the GWAS index SNPs for both the number of observations and minor-allele-frequency. In particular, for each GWAS index SNP we generated two sets (set I, set II) of matching SNPs that (a) excluded the six GWAS index SNPs, (b) had number of inferred Central Andean and Mapuche haplotypes within 20 of that for the index SNP, and (c) had minor-allele-frequency within 1% of the index SNP among inferred Central Andean haplotypes (set I) or inferred Mapuche haplotypes (set II). The matching SNP counts for each GWAS index SNP in each of set I and set II, plus the empirical p-values, are provided in Supplementary Table 6.

**Reporting summary**. Further information on research design is available in the Nature Research Reporting Summary linked to this article.

**Code availability**. SOURCEFIND is available at www.paintmychromosomes.com.

## Data availability

Raw genotype or phenotype data cannot be made available due to restrictions imposed by the ethics approval. Summary statistics from previous GWAS on the CANDELA consortium data have been deposited in GWAS central [https://www.gwascentral.org/study/HGVST1841/, http://www.gwascentral.org/study/HGVST3308]. The publicly available reference population datasets were obtained from: (1) Mallick et al.[48]: EBI ENA PRJEB9586; (2) Eichstaedt et al.[47]: NCBI GEO GSE55175; (3) 1000 Genomes Project[46] [http://www.internationalgenome.org/data]; (4) Pagani et al.[49]: [http://mega.bioanth.cam.ac.uk/data/Ethiopia/] and (5) Schlebusch et al.[50]: [http://jakobssonlab.iob.uu.se/data/].

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

## Acknowledgements

We would like to dedicate this paper to the memory of L.L. Cavalli-Sforza, a wonderful mentor and a constant source of inspiration. We are grateful to the volunteers for supporting this research. We thank Alvaro Alvarado, William Arias, Mónica Ballesteros Romero, Ricardo Cebrecos, Miguel Ángel Contreras Sieck, Francisco de Ávila Becerril, Joyce De la Piedra, María Teresa Del Solar, Gastón Macín, William Flores, Martha Granados Riveros, Rosilene Paim, Ricardo Gunski, Sergeant João Felisberto Menezes Cavalheiro, Major Eugênio Correa de Souza Junior, Wendy Hart, Ilich Jafet Moreno, Claudia Jaramillo, Paola León-Mimila, Francisco Quispealaya, Diana Rogel Diaz, Ruth Rojas and Vanessa Sarabia, for assistance with volunteer recruitment, sample processing and data entry. We acknowledge the institutions that provided facilities for the assessment of volunteers: Escuela Nacional de Antropología e Historia and Universidad Nacional Autónoma de México (México); Universidade Federal do Rio Grande do Sul (Brazil); 13° Companhia de Comunicações Mecanizada do Exército Brasileiro (Brazil); Pontificia Universidad Católica del Perú, Universidad de Lima and Universidad Nacional Mayor de San Marcos (Perú). We also thank the National Laboratory for the Genetics of Israeli Populations (http://yoran.tau.ac.il/nlgip/) and Dr. David Gurwitz for making available DNA samples. We thank Chris Tyler-Smith and Caroline Costedoat for comments on the manuscript. Work leading to this publication was funded by grants from: the Leverhulme Trust (F/07 134/DF), BBSRC (BB/I021213/1), the Excellence Initiative of Aix-Marseille University—A*MIDEX (a French Investissements d'Avenir programme), Wellcome Trust/Royal Society (098386/Z/12/Z to G.H.), Universidad de Antioquia (CODI sostenibilidad de grupos 2013–2014 and MASO 2013–2014), Conselho Nacional de Desenvolvimento Científico e Tecnológico, Fundação de Amparo à Pesquisa do Estado do Rio Grande do Sul (Apoio a Núcleos de Excelência Program) and Fundação de Aperfeiçoamento de Pessoal de Nível Superior. V.G. is supported by Fundação para a Ciência e Tecnologia (FCT) and Programa Operacional Potencial Humano (POCH), through the grant SFRH/BPD/76207/2011. IPATIMUP integrates the i3S Research Unit, which is partially supported by FCT. Y.X. was supported by The Wellcome Trust (098051). K.A. was supported by an UCL Global Engagement Fund, and a Wellcome Investigator Award WT107055AIA (to Prof. C.D. Stern). J.C.C.D. was supported by a doctoral scholarship from COLCIENCIAS (Colombia).

## Author contributions

J.C.C.D., K.A., J.M.R., M.F.G., G.H. and A.R.L. performed the analyses. G.H. developed SOURCEFIND. J.C.C.D., K.A., G.H. and A.R.L. wrote the paper with input from co-authors. D.B. and R.G.-J. provided advice on study design and statistical analysis. All other authors, namely, A.A., A.S.-G., C.B., C.G., C.R., C.S.D.C., C.W., D.C., E.L., F.R., F. S., G.B., G.P., H.R.-V., H.V.-R., J.G.-V., J.M., J.-M.D., J.S., L.G., L.S.-F., M.-C.B., M.H., M.-L.P., M.Q.-S., M.V., P.E.M., P.M., P.P.-E., R.B., R.B.L., R.V., S.C.-Q., T.H., V.A.-A., V. Go., V.Gr., V.R., V.V., W.K., and Y.X., contributed to volunteer recruitment or collection of data. A.R.L. coordinated the study.

## Additional information

**Competing interests:** The authors declare no competing interests. G.H. is a founding member of GenSci. J.C.C.D was employed by Living DNA from October 2017 to November 2018.

Juan-Camilo Chacón-Duque [1], Kaustubh Adhikari [1], Macarena Fuentes-Guajardo[1,2], Javier Mendoza-Revilla[1,3], Victor Acuña-Alonzo[1,4], Rodrigo Barquera [4,5], Mirsha Quinto-Sánchez[6], Jorge Gómez-Valdés [7], Paola Everardo Martínez[8], Hugo Villamil-Ramírez[9], Tábita Hünemeier[10], Virginia Ramallo[11,12], Caio C. Silva de Cerqueira[12,39], Malena Hurtado[3], Valeria Villegas[3], Vanessa Granja[3], Mercedes Villena[13], René Vásquez[14], Elena Llop[15], José R. Sandoval [16], Alberto A. Salazar-Granara[16], Maria-Laura Parolin[17], Karla Sandoval[18], Rosenda I. Peñaloza-Espinosa[19], Hector Rangel-Villalobos[20], Cheryl A. Winkler[21], William Klitz[22], Claudio Bravi[23], Julio Molina[24], Daniel Corach[25], Ramiro Barrantes[26], Verónica Gomes [27,28], Carlos Resende[27,28], Leonor Gusmão[27,28,29], Antonio Amorim [27,28,30], Yali Xue[31], Jean-Michel Dugoujon[32], Pedro Moral[33], Rolando González-José[11], Lavinia Schuler-Faccini[12], Francisco M. Salzano[12], Maria-Cátira Bortolini[12], Samuel Canizales-Quinteros[9], Giovanni Poletti[3], Carla Gallo[3], Gabriel Bedoya[34], Francisco Rothhammer[35,15], David Balding [1,36], Garrett Hellenthal[1] & Andrés Ruiz-Linares[37,38]

[1]Department of Genetics, Evolution and Environment and UCL Genetics Institute, University College London, London WC1E 6BT, UK. [2]Departamento de Tecnología Médica, Facultad de Ciencias de la Salud, Universidad de Tarapacá, Arica 1000009, Chile. [3]Laboratorios de Investigación y Desarrollo, Facultad de Ciencias y Filosofía, Universidad Peruana Cayetano Heredia, Lima 31, Peru. [4]Molecular Genetics Laboratory, Escuela Nacional de Antropología e Historia, Mexico City 14030, Mexico. [5]Department of Archaeogenetics, Max Planck Institute for the Science of Human History, Jena 07745, Germany. [6]Ciencia Forense, Facultad de Medicina, Universidad Nacional Autónoma de México, Mexico City 04510, Mexico. [7]Posgrado en Antropología Física, Escuela Nacional de Antropología e Historia, Mexico City 14030, Mexico. [8]Posgrado en Antropología, Universidad Nacional Autónoma de México, Mexico City 04510, Mexico. [9]Unidad de Genómica de Poblaciones Aplicada a la Salud, Facultad de Química, Universidad Nacional Autónoma de México e Instituto Nacional de Medicina Genómica, Mexico City 04510, Mexico. [10]Departamento de Genética e Biología Evolutiva, Instituto de Biociências, Universidade de São Paulo, Sao Paulo 05508-090, Brazil. [11]Instituto Patagónico de Ciencias Sociales y Humanas-Centro Nacional Patagónico, CONICET, Puerto Madryn U912OACD, Argentina. [12]Departamento de Genética, Universidade Federal do Rio Grande do Sul, Porto Alegre 91501-970, Brazil. [13]Instituto Boliviano de Biología de Altura (IBBA), Universidad Mayor de San Andrés (UMSA), La Paz 2070, Bolivia. [14]Instituto Boliviano de Biología de Altura (IBBA), Universidad Autónoma Tomás Frías, Potosí 53820, Bolivia. [15]Programa de Genetica Humana, ICBM, Facultad de Medicina, Universidad de Chile, Santiago 1027, Chile. [16]Facultad de Medicina Humana, Universidad de San Martín de Porres, Lima 12, Peru. [17]Instituto de Diversidad y Evolución Austral (IDEAus), Centro Nacional Patagónico, CONICET, Puerto Madryn U912OACD, Argentina. [18]National Laboratory of Genomics and Biodiversity (LANGEBIO), CINVESTAV, Irapuato 36821, Mexico. [19]Department of Biological Systems, Division of Biological and Health Sciences, Universidad Autónoma Metropolitana-Xochimilco, Mexico City 04960, Mexico. [20]Instituto de Investigación en Genética Molecular, Universidad de Guadalajara, Ocotlán 1115, Mexico. [21]Basic Research Laboratory, National Cancer Institute, Frederick National Laboratory, Frederick, MD 21702, USA. [22]Integrative Biology, University of California, Berkeley, CA 94720, USA. [23]Instituto Multidisciplinario de Biología Celular, CONICET, La Plata B1906APO, Argentina. [24]Centro de Investigaciones Biomédicas de Guatemala, Ciudad de Guatemala 01011, Guatemala. [25]Servicio de Huellas Digitales Genéticas and CONICET, Universidad de Buenos Aires, Buenos Aires C1113AAD, Argentina. [26]Escuela de Biología, Universidad de Costa Rica, San José 2060, Costa Rica. [27]Instituto de Patologia e Imunologia Molecular da Universidade do Porto (IPATIMUP), Porto 4200-135, Portugal. [28]Instituto de Investigação e Inovação em Saúde (i3S), Universidade do Porto, Porto 4200-135, Portugal. [29]DNA Diagnostic Laboratory (LDD), Universidade do Estado do Rio de Janeiro, Rio de Janeiro 23968-000, Brazil. [30]Faculdade de Ciências, Universidade do Porto, Porto 4169-007, Portugal. [31]The Wellcome Trust Sanger Institute, Hinxton CB10 1SA, UK. [32]Centre National de la Recherche Scientifique, Université Toulouse 3 Paul Sabatier, Toulouse 31330, France. [33]Departamento de Biología Evolutiva, Ecología y Ciencias Ambientales, Universitat de Barcelona, Barcelona 08007, Spain. [34]Genética Molecular (GENMOL), Universidad de Antioquia, Medellín 5001000, Colombia. [35]Instituto de Alta Investigación, Universidad de Tarapacá, Arica 1000009, Chile. [36]Schools of BioSciences and Mathematics & Statistics, University of Melbourne, Melbourne, VIC 3010, Australia. [37]Ministry of Education Key Laboratory of Contemporary Anthropology and Collaborative Innovation Center of Genetics and Development, School of Life Sciences and Human Phenome Institute, Fudan University, Shanghai 200433, China. [38]Aix-Marseille Univ, CNRS, EFS, ADES, Marseille 13007, France. [39]Present address: Scientific Police of São Paulo State, Ourinhos-SP 19900-109, Brazil. Deceased: Francisco M. Salzano. These authors jointly supervised: Garrett Hellenthal, Andrés Ruiz-Linares.

