## [Peer Review File · Nature Communications]

Reviewer #2 (Remarks to the Author):

1) They used the Latin American populations with well-known historical knowledge of migration in the last 500~ years. It is rare to find this kind of large scale native and immigrant population admixtures in the world. Therefore, it is an interesting genomic resource. These authors have published papers using CANDELA set.

2) The detailed and high definition analysis of Latin American genomics has some recent programs and a new algorithm called SOURCEFIND.

This author finds this an interesting "resource" to population genomics. The overall scientific discovery impact is small.

It is a short and clear manuscript with not many novel findings.

However, it is a good survey.

1) What are small dark dots in Figure 1 ?

2) Is it required to have a contents in Supp and put page numbers instead of putting Table and Figure numbers on the tables and Figures? (Not easy to navigate the supp)

3) "We find that pre-Columbian Native genetic structure is mirrored in Latin Americans and that sources of non-Native ancestry, and admixture timings, match documented migratory flows."

This review thinks that the above is perhaps the most important finding of the analyses. It is not surprising. Does that just simply reflect HW equilibrium?

4) "little is known about the genetic basis of variation in physical appearance within continental human populations."... "its impact on physical appearance."

I agree. However, within a continent or a region, it should not be too distinct unless there is some social or artificial reasons.

The authors suggest that they could provide fine-detailed evidence for such variation. However, this reviewer cannot see it clearly.

5) "We grouped the reference population individuals into 56 homogeneous clusters based on patterns of haplotype sharing, using the program fineSTRUCTURE2 "

Did the authors a priori designate the number 56 ? Or did fineStructure came up with 56 references? Should be stated in anyway in the main text. Overall, defining the 56 seems quite a big task with a lengthy explanation. It is not easy to understand it.

...."In order to reduce the number of clusters potentially representing sources of ancestry in Latin America, to avoid problems related to collinearity between different surrogate sources when estimating ancestry, and to facilitate interpretation of the results, we carried out the refinements described below, leading to the re-assignment of individuals from these 129 clusters into 117 "donor clusters". "....

6) SOURCEFIND is essentially the core of this manuscript and analyses. Is it so much of an improvement over structure, admixture and other components clustering/finding programs?(It is not clear to me. It would have been good if it is stated or clear that this new program is 10% more accurate at a definable resolution etc.)

"The accuracy and robustness of the ancestry estimations obtained by SOURCEFIND and NNLS were evaluated using simulations mimicking Latin American admixture (Supplementary Note 1)."

7) "Therefore, for some analyses we used an alternative, more computationally efficient version of SOURCEFIND that uses the same likelihood function, but which removes Lambda and replaces the prior on the βr values with a truncated Poisson (mean=3) pri-or on the number of contributing surrogates S'. At each MCMC iteration, this alternative SOURCEFIND allows only a maximum of S' surrogates to have $\beta sr > 0$ and for the βsr values of each of these S' surrogates to be 0.01,...,1 in increments of 0.01." ...

How expensive is it?(sourcefind). As the introducing manuscript, wouldn't it be necessary to show comparative numbers on the expensive and non-expensive version?

Overall, this is a large-scale in-depth analyses of a big set of populations using a new program. I am not an expert on the algorithm(clustering) parts and cannot see the impact of it in our population genomics. However, it is not clear to me what scientifically notable aspects are presented as a citable paper apart from CANDELA set.

Reviewer #3 (Remarks to the Author):

The manuscript by Chacon-Duque provides a detailed analysis of ancestry in a variety of Latin American populations. Specifically, they develop a new method, called Sourcefind, which allows local ancestry in admixed individuals to be assigned to specific ancestral populations. The authors make a number of interesting findings, including that there is South/East Mediterranean ancestry in Latin American populations. This could be due to Jewish immigrants fleeing Europe. Lastly, the authors find some associations between ancestry and certain physical traits.

Overall, I found this manuscript to be quite exciting, and contains many novel, interesting, and important findings. However, I have a number of concerns about the manuscript which I outline below:

1) Abstract: I found this to be a bit vague in certain important places. For example, the statement: "evaluate the impact of sub-continental ancestry on the physical appearance of these individuals" should be made more precise. The last phrase of the abstract, "possibly relating to environmental

adaptation during the evolution of Native Americans” was unduly speculative. I also found the discussion of the nasal cavity association in the second to last paragraph of the main text to be speculative and the weakest part of the paper. Unless there can be a more direct link made about whether this trait could be due to selection, adaptation shouldn’t be mentioned in the abstract, and should either be removed from the main text, or discussed in a much more balanced way.

2) Main text: Overall, I found the main text to be quite dense and terse in certain spots. I think the paper would greatly benefit from expanding the explanation of certain tests and issues. For example, in Figure 3, the authors are testing whether the timings of admixture differ between different ancestry components from Iberia. The specific rationale for this analysis should be better outlined in the main text. Why is everything being compared to Iberia? What are the specific hypotheses being tested? This should be more clearly articulated. Additionally, the Converso migration is mentioned in the title as one of the main findings. Yet, there are only a few sentences on this in the main text. Some additional discussion of this would strengthened the paper and make it more readable.

3) At a number of points in the paper, the authors try to connect specific genetic signatures that they find to certain historical events. Obviously, this is an interesting and important thing to do. However, I think there should be more discussion about other possible explanations for these patterns in the data. Specifically, could there be other explanations for the “signature of the Converso’ migration?

More minor comments:

1) Page 10, line 449: The writing would be made clearer if M was defined earlier on.

2) Page 10, line 465 and 474: The proportional sign should be changed to the Greek letter alpha.

3) Page 11, line 517: Is 2378 a typo? Shouldn’t it be 6352?

Reviewer #4 (Remarks to the Author):

This study is an in-depth analysis of the fine population structure of Latin America, using a large sample size covering many potential source populations. The patterns found in some cases match documented migrations, and also explain some phenotypic differences across populations. The researchers also find signatures of admixture that reflect migrations that were poorly documented for political and societal reasons (in the form of Sephardic Jews escaping persecution in Iberia), which is extremely neat.

General Comments:

1. What do you think the effect would be on this type of study of missing source populations? Granted, this study uses a very large number of populations as potential sources, but some discussion on this would be generally useful. For example, if a true source population is not represented, but a closely related population is instead, might that have an affect on the proportions of ancestry calculated, or on the estimated times of admixture?
2. In a couple of places (line 156, 202), ongoing admixture, or admixture over multiple generations, is invoked to explain more recent inferred dates of admixture. It would be useful to have a discussion of what the different signatures are expected to be of a single pulse of admixture v. admixture over multiple generations. For example, if admixture is ongoing, how does that skew estimates of admixture time inferred by GLOBETROTTER? Either a discussion with citations, or an illustration using simulations would be good.
3. All of the Wilcoxon rank-sum tests are one-sided, but there is no discussion of why all other admixture events are expected to be more recent than the Iberian admixture. This should go somewhere.
4. I found the explanation of the pipeline in the methods extremely confusing (the use of CHROMOPAINTER, NNLS, fineSTRUCTURE, and SOURCEFIND, and the definition of all of the different sets of population clusters) — a schematic figure (probably in the supplement) would be extremely useful here. In particular, it is a bit unclear why you use both NNLS and SOURCEFIND, and what they are doing differently. I gather that NNLS is used as a preliminary step because it is more computationally efficient? This should be made much more clear upfront to help orient the reader to what each step of the pipeline is doing.
5. Ideally, the SOURCEFIND software should be made publicly available as a github repository (e.g.) to better facilitate reproduction of these results, as well as the furthering of other studies.

Specific Comments:

1. line 125: "simulations show that SOURCEFIND has greater accuracy than other approaches..." It would be helpful to name them here. By the supplement, it looks like the only comparison is to NNLS. Are there others that do similar tasks? If so, some brief discussion of those would be appropriate. Also, from what I understand, although it is more accurate, SOURCEFIND is more computationally expensive — it would be good to mention that here or elsewhere in the main text, especially in the context of the analysis pipeline.

2. line 192: "arguably with a peculiar history" — what does this mean? I'm intrigued, but confused.

3. line 195-196: I'm sort of confused by this sentence. Would you expect Sub-Saharan ancestry to be higher than 4%? Why are regions that received large numbers of African slaves under-represented?

4. line 206-208: Does more Chinese ancestry in the non-Brazilian CANDELA individuals, and more Japanese ancestry in the Brazilian individuals match any migration documentation? Or could this be an artifact of the Brazilian individuals having different distributions of other ancestry components?

Below, our replies to the Reviewers' comments are shown in bold.

Reviewer #2:

They used the Latin American populations with well-known historical knowledge of migration in the last 500~ years. It is rare to find this kind of large scale native and immigrant population admixtures in the world. Therefore, it is an interesting genomic resource. These authors have published papers using CANDELA set.

The detailed and high definition analysis of Latin American genomics has some recent programs and a new algorithm called SOURCEFIND.

This author finds this an interesting "resource" to population genomics. The overall scientific discovery impact is small.

It is a short and clear manuscript with not many novel findings.

We respectfully disagree with this reviewer. Our paper includes a number of novel findings, as highlighted by the other two reviewers. We believe the media interest (<http://www.sciencemag.org/news/2018/04/latin-america-s-lost-histories-revealed-modern-dna>) and twitter comments (<https://twitter.com/cdbustamante/status/956171314678571009>, <https://www.gnpx.com/WordPress/2018/01/24/the-conversos-in-the-spanish-empire-and-undoing-anthropological-mythologists/>) based on the bioRxiv early version of the manuscript (<https://www.biorxiv.org/content/early/2018/01/23/252155>) also show our paper is of wide-ranging scientific (and public) interest.

However, it is a good survey.

1) What are small dark dots in Figure 1?

The reviewer likely refers to the grey dots, which we describe in the figure legend: "*The grey dots indicate reference populations not inferred to contribute ancestry to the CANDELA sample*". We understand that the colour and size of the dots can cause confusion, so we have modified the text to say "small dark grey dots".

2) Is it required to have a contents in Supp and put page numbers instead of putting putting Table and Figure numbers on the tables and Figures? (Not easy to navigate the supp)

The table of Contents at the beginning of the Supplementary Material lists the page numbers for each section and also for each labelled Table and Figure.

3) "We find that pre-Columbian Native genetic structure is mirrored in Latin Americans and that sources of non-Native ancestry, and admixture timings, match documented migratory flows."

This review thinks that the above is perhaps the most important finding of the analyses. It is not surprising. Does that just simply reflect HW equilibrium?

We find this comment puzzling. Hardy Weinberg equilibrium (HWE) describes how allele frequencies remain constant in a population from generation to generation, so it is unclear how this comment relates to the findings on Latin American genetic structure that we report here. Population sub-division and admixture are violations of the assumptions of HWE, so our results – which strongly support both – indicate that HWE is violated, though that may depend on how you define a "population". For such reasons, we feel that commenting on HWE will confuse readers.

4) "little is known about the genetic basis of variation in physical appearance within continental human populations."... "its impact on physical appearance."

I agree. However, within a continent or a region, it should not be too distinct unless there is some social or artificial reasons.

The authors suggest that they could provide fine-detailed evidence for such variation. However, this reviewer cannot see it clearly.

We disagree with the reviewer on this point. Although “little is known about the genetic basis of variation in physical appearance within continental populations”, morphological studies have extensively demonstrated that many physical appearance traits vary between populations (see references in the paper) and in particular among populations from the countries we examined here (Comas, J. Antropologia de los pueblos iberoamericanos. Biblioteca Universitaria Labor, 1974). The causes for this variation may be environmental (including social), as the reviewer proposes, but they certainly exist.

In this paper we show how sub-continental ancestry is associated with this previously observed variation in physical appearance across Latin America. Specifically, we demonstrate that variation in Native sub-continental ancestry in the Andean region significantly impacts on facial features such as nose morphology, and that variation in Northern versus Southern European ancestry significantly impacts on pigmentation phenotypes among Brazilians. We also provide additional evidence by finding statistically significant differences in allele frequencies between Central Andeans and Mapuches in loci associated with variation in facial traits. We have now included a new analysis that further shows the allele frequencies at these loci are unusually differentiated between Central Andeans and Mapuches relative to genome-wide average, suggesting they have been subjected to selection. Hopefully, the fact that this reviewer, *a priori*, did not believe such associations should exist helps to showcase one of the novelties of this paper.

5) "We grouped the reference population individuals into 56 homogeneous clusters based on patterns of haplotype sharing, using the program fineSTRUCTURE2 "

Did the authors a priori designate the number 56 ? Or did fineStructure came up with 56 references? Should be stated in anyway in the main text. Overall, defining the 56 seems quite a big task with a lengthy explanation. It is not easy to understand it.

...."In order to reduce the number of clusters potentially representing sources of ancestry in Latin America, to avoid problems related to collinearity between different surrogate sources when estimating ancestry, and to facilitate interpretation of the results, we carried out the refinements described below, leading to the re-assignment of individuals from these 129 clusters into 117 “donor clusters”. "....

We acknowledge the reviewer's concerns about the necessity of more clarity in the definition of the clusters, and we have made an effort to explain it more clearly in the methods of the main text. Briefly, the 56 was not decided a priori but was based on the fineSTRUCTURE clustering, which gave 129 clusters, after which some clusters were selected to represent the sources of ancestry of Latin Americans using a series of additional analyses, to avoid potential complications with the analyses and to assist interpretation of results. Reviewer #3 also raised this issue and suggested to include a schematic figure in the supplement explaining the whole clustering process. We now have included such a figure (Supplementary Fig. 13), and we hope that the changes in the text and the schematic figure improve the clarity of the processes description.

6) SOURCEFIND is essentially the core of this manuscript and analyses. Is it so much of an improvement over structure, admixture and other components clustering/finding programs?(It is not clear to me. It would have been good if it is stated or clear that this new program is 10% more accurate at a definable resolution etc.)

"The accuracy and robustness of the ancestry estimations obtained by SOURCEFIND and NNLS were evaluated using simulations mimicking Latin American admixture (Supplementary Note 1)."

Based on the reviewer's suggestion, we have expanded the explanation regarding this improvement. Firstly, we added more information in the main text (first paragraph on results), making clear that the comparisons on the simulated data show a marked improvement of SOURCEFIND compared to NNLS. We note it is challenging to make a general statement about percentage improvement, as this is context dependent and involves both up-weighting the inferred contributions of the correct sources and down-weighting the inferred contributions of incorrect sources. We hope the first four figures of Supplementary Note 2 show SOURCEFIND's improvements over NNLS quite clearly, largely in terms of eliminating inferred contributions from the wrong source groups. Secondly, we have also added in the same paragraph a note about how ADMIXTURE analyses in the CANDELA sample show clear limitations when estimating sub-continental ancestry, which we detail in Supplementary Note 1: "Analysis of these data using the allele-frequency-based approach ADMIXTURE (Alexander, 2009) show major limitations for estimating sub-continental ancestry (Supplementary Note 3), similar to what is seen in other datasets (Lawson, 2017; Lawson, 2012)"

7) "Therefore, for some analyses we used an alternative, more computationally efficient version of SOURCEFIND that uses the same likelihood function, but which removes Lambda and replaces the prior on the βr values with a truncated Poisson (mean=3) pri-or on the number of contributing surrogates S' . At each MCMC iteration, this alternative SOURCEFIND allows only a maximum of S' surrogates to have $\beta sr > 0$ and for the βsr values of each of these S' surrogates to be 0.01,...,1 in increments of 0.01." ...

How expensive is it?(sourcefind). As the introducing manuscript, wouldn't it be necessary to show comparative numbers on the expensive and non-expensive version?

In the supplementary material (2nd paragraph of Supplementary Note 2) we have now explicitly compared the run times, while also providing a direct comparison of both SOURCEFIND versions on the simulated data:

“In particular in each of simulations (i)-(iv) described below, we provide plots illustrating the accuracy of both the initial SOURCEFIND version (called “SOURCEFIND1” in this section) and the computationally efficient version of SOURCEFIND (called “SOURCEFIND2”). For these simulations, accuracy is only very slightly reduced when using SOURCEFIND2 relative to SOURCEFIND1. Regarding computation time, analysis of a single CANDELA individual took ~10 minutes using each run of the initial SOURCEFIND version with 200K MCMC iterations, hence taking $10 \times 50 = 500$ minutes to do 50 independent runs. In contrast, it took ~25 seconds with 100K MCMC iterations and a single run of the more computationally efficient version. We note that additional independent runs of the computationally efficient version may improve performance, while reducing the gains in computation time (e.g. doing 50 independent runs would make it only ~20x faster than the initial SOURCEFIND). ”

Overall, this is a large-scale in-depth analyses of a big set of populations using a new program. I am not an expert on the algorithm(clustering) parts and cannot see the impact of it in our population genomics. However, it is not clear to me what scientifically notable aspects are presented as a citable paper apart from CANDELA set.

We disagree for the reasons noted above. We also note that researchers from 6 universities have already requested the new SOURCEFIND technique introduced in this paper after seeing the archived version.

Reviewer #3:

The manuscript by Chacon-Duque provides a detailed analysis of ancestry in a variety of Latin American populations. Specifically, they develop a new method, called Sourcefind, which allows local ancestry in admixed individuals to be assigned to specific ancestral populations. The authors make a number of interesting findings, including that there is South/East Mediterranean ancestry in Latin American populations. This could be due to Jewish immigrants fleeing Europe. Lastly, the authors find some associations between ancestry and certain physical traits.

Overall, I found this manuscript to be quite exciting, and contains many novel, interesting, and important findings. However, I have a number of concerns about the manuscript which I outline below:

1) Abstract: I found this to be a bit vague in certain important places. For example, the statement: “evaluate the impact of sub-continental ancestry on the physical appearance of these individuals” should be made more precise.

We agree that this sentence in the abstract sounded vague, given the limited size of this section, we have rewritten this sentence to make it clearer. We have changed the sentence to: “Using novel haplotype-based methods, here we infer sub-continental ancestry in over 6,500 Latin Americans and evaluate the impact of regional ancestry variation on physical appearance”.

The last phrase of the abstract, “possibly relating to environmental adaptation during the evolution of Native Americans” was unduly speculative. I also found the discussion of the nasal cavity association in the second to last paragraph of the main text to be speculative and the weakest part of the paper. Unless there can be a more direct link made about whether this trait could be due to selection, adaptation shouldn’t be mentioned in the abstract, and should either be removed from the main text, or discussed in a much more balanced way.

We agree that the adaptation argument sounded speculative. We have now explicitly tested whether the allele frequencies at loci found in a previous GWAS study to be associated with facial features are more highly differentiated than typical between haplotypes inferred to be of Central Andean ancestry versus Mapuche ancestry, which is indicative of selection at these loci. We have adjusted the following paragraph in the main text that the reviewer refers to:

“The Mapuche component is strongly associated with a less protruded nose (P-value $<2 \times 10^{-5}$) and broader nose tip angle (P-value $<10^{-7}$). This is consistent with physical anthropology studies indicating that the Mapuche have a flatter, wider nose than Central Andean populations (Comas, 1974). In a recent genome-wide association scan (GWAS) for facial features in the CANDELA sample, most loci identified impacted on nose shape (Adhikari et al, 2016). For each of the six reported SNPs significantly associated with facial features in that GWAS, allele frequencies at haplotypes inferred to be of Central Andean ancestry were significantly different from allele frequencies at haplotypes inferred to be of Mapuche ancestry (Supplementary Table 5). Furthermore, for each of these six, the frequency of the allele associated with an increase of the phenotypic trait was higher in the Native component associated with an increase of that trait. The nasal cavity is an important regulator of inhaled air temperature and humidity, and evolutionary studies suggest that nose shape has been influenced by adaptation to cold/dry versus hot/humid environments (Zaidi et al, 2017). Consistent with selection effects at these SNPs, allele frequencies at the set of six GWAS hit SNPs jointly were more differentiated between Central Andean and Mapuche than was the case in randomly selected sets of six genome-wide SNPs that matched each

hit SNP for the number of inferred Native ancestry haplotypes and minor allele frequency in either the inferred Central Andean (p-value < 0.02) or inferred Mapuche (p-value < 0.01) haplotypes (Supplementary Figures 10 and 11, Supplementary Table 6). Since variation in altitude correlates with air temperature and humidity, it will be interesting to explore further whether the association of Central Andean ancestry with nose shape and the differences in allele frequencies in loci associated with facial features between Central Andeans and the Mapuche relate to altitude adaptation during Native American evolution.”

Our selection test procedure is described in the Methods, with new figures and tables (mentioned in the above text) added to the Supplementary Material:

“We also assessed whether the allele frequencies at these six SNPs jointly were excessively differentiated between haplotypes inferred to be of Central Andean ancestry versus those inferred to be of Mapuche ancestry. To do so, we randomly selected sets of six genome-wide SNPs. For each SNP in the set of six, we used the same t-test to calculate a p-value testing the null hypothesis that the Central Andean and Mapuche allele frequencies were the same, taking the average $\log_{10}(\text{p-value})$ across all six SNPs in the set. We found the proportion of 10,000 such random samples of six SNPs with average $\log_{10}(\text{p-value})$ less than or equal to that of the six GWAS hit SNPs, using this proportion as an empirical p-value testing whether the six GWAS hit SNPs were more differentiated than usual. In order to match power between our six GWAS hit SNPs and each random set of six SNPs, we only randomly selected from SNPs matched to the GWAS hit SNPs for both the number of observations and minor-allele-frequency. In particular, for each GWAS hit SNP we generated two sets (“set I”, “set II”) of matching SNPs that (a) excluded the six GWAS hit SNPs, (b) had number of inferred Central Andean and Mapuche haplotypes within 20 of that for the hit SNP, and (c) had minor-allele-frequency within 1% of the hit SNP among inferred Central Andean haplotypes (“set I”) or inferred Mapuche haplotypes (“set II”). The matching SNP counts for each GWAS hit SNP in each of “set I” and “set II”, plus the empirical p-values, are provided in Supplementary Table 6.”

Finally, following the reviewer's suggestion, we have now changed the last sentence of the abstract to: "Furthermore, we find that ancestry related to highland (Central Andean) versus lowland (Mapuche) Natives is associated with variation in facial features, particularly nose morphology, and detect significant differences in allele frequencies between these groups, indicative of selection, at loci previously associated with nose morphology in this sample."

2) Main text: Overall, I found the main text to be quite dense and terse in certain spots. I think the paper would greatly benefit from expanding the explanation of certain tests and issues. For example, in Figure 3, the authors are testing whether the timings of admixture differ between different ancestry components from Iberia. The specific rationale for this analysis should be better outlined in the main text. Why is everything being compared to Iberia? What are the specific hypotheses being tested? This should be more clearly articulated.

Overall we have substantially increased the text to address this point, hopefully adding more clarity. Regarding the specific point about the use of Iberia as a comparison point, we did so because Iberians are believed to be the first colonizers into the Americas, so that other admixture events should have more recent dates if not related to these initial colonial migrations. We have updated the main text to clarify why we compare to Iberian dates by noting how our findings match up to this known history throughout the main text:

"Inferred dates for events involving an Iberian source (the initial settlers arriving from Europe and allegedly the first to admix with the natives) had a median of 10 generations (IQR=7-13), consistent with other estimates for admixture in Latin America (Wang et al. 2008, Moreno-Estrada et al. 2013, Homburger et al. 2015).

"Compared to inferred dates related to Iberian admixture, admixture events involving non-Iberian European sources (Northwest Europe, Italy) have a significant skew towards more recent dates (Fig. 3B; Wilcoxon rank-sum test one-sided p -value= 3×10^{-8}) (Kent 2016), consistent with the relatively recent arrival of Germans and Italians"

"...Furthermore, GLOBETROTTER estimates for the time since East/South Mediterranean admixture were not significantly different from those involving Iberian sources (Fig. 3C; Wilcoxon rank-sum test one-sided p -value >0.1), consistent with most of this ancestry component being contributed simultaneously with the initial colonial immigrants..."

".... The distribution of dates involving Sub-Saharan African admixture mostly overlaps with that for Iberian admixture, although a substantial proportion of recent dates were also inferred (Fig. 3D), possibly reflecting continuing African admixture in the regions sampled."

"Reflecting the relatively recent nature of these events, GLOBETROTTER estimated dates for admixture involving an East Asian source were significantly more recent than those involving Iberian sources (median = 3, IQR 2-5 generations ago, Wilcoxon rank-sum test one-sided p -value $<10^{-15}$; Fig. 3E)."

Additionally, the Converso migration is mentioned in the title as one of the main findings. Yet, there are only a few sentences on this in the main text. Some additional discussion of this would strengthened the paper and make it more readable.

We totally agree that given the fact this is one of the most important findings on the paper, it deserves some additional discussion, and we have considerably revised and expanded the paragraph.

3) At a number of points in the paper, the authors try to connect specific genetic signatures that they find to certain historical events. Obviously, this is an interesting and important thing to do. However, I think there should be more discussion about other possible explanations for these patterns in the data. Specifically, could there be other explanations for the "signature of the Converso' migration?"

We now address the Converso scenario directly in the Discussion, listing additional potential caveats in our analyses as well. For example, we have added:

"A further complication is that some of the reference populations may have experienced admixture following the colonial period. For example, it is possible that the Iberian reference individuals examined here have less non-European (East/South Mediterranean and/or Sub-Saharan African) ancestry than individuals migrating to the Americas during the colonial period due to ongoing admixture with other Europeans. In this case SOURCEFIND may overestimate the contributions from the non-European groups. Because of this, estimates for each of the East/South /Mediterranean and African components should be interpreted as values over and above those present in the Spanish/Portuguese reference individuals examined. As noted above, the similarity in inferred dates for admixture involving East/South Mediterranean versus Iberian ancestry furthermore suggests that the individuals carrying this excess East/South Mediterranean ancestry migrated to Latin America during the colonial period."

More minor comments:

1) Page 10, line 449: The writing would be made clearer if M was defined earlier on.

This has been clarified by adding the definition in the same line:

“... Then we perform the following for $m \in [1, \dots, M]$, where M is the total number of MCMC iterations: ”

2) Page 10, line 465 and 474: The proportional sign should be changed to the Greek letter alpha.

Thanks for spotting this -- we have corrected it.

3) Page 11, line 517: Is 2378 a typo? Shouldn't it be 6352?

Thanks for spotting this – it has been corrected.

Reviewer #4:

This study is an in-depth analysis of the fine population structure of Latin America, using a large sample size covering many potential source populations. The patterns found in some cases match documented migrations, and also explain some phenotypic differences across populations. The researchers also find signatures of admixture that reflect migrations that were poorly documented for political and societal reasons (in the form of Sephardic Jews escaping persecution in Iberia), which is extremely neat.

General Comments:

1. What do you think the effect would be on this type of study of missing source populations? Granted, this study uses a very large number of populations as potential sources, but some discussion on this would be generally useful. For example, if a true source population is not represented, but a closely related population is instead, might that have an effect on the proportions of ancestry calculated, or on the estimated times of admixture?

This is certainly an important point. Briefly, previous work applying these approaches to simulations that remove the true admixing sources prior to performing inference suggest that the inferred dates are robust, while the inferred proportions can vary substantially (Hellenthal et al 2014). We have included detailed comments on this in the Discussion:

"While previous work has suggested GLOBETROTTER's inferred dates are robust to using different surrogates to the true ancestry sources (Hellenthal et al 2014), inferred proportions of ancestry inevitably depend on which surrogate groups are used. In general our SOURCEFIND inference suggests that the reference populations included in this study are good representatives of the true ancestral sources, as demonstrated by the preferential matching to specific geographic regions of Iberia (Fig 2B) and the strong correspondence between geography and ancestry matching in the Native component (Fig 2A). A caveat to this is that some of our reference Native groups evidenced strong genetic drift and SOURCEFIND inferred negligible contributions from such groups (Supplementary Table 7). Indeed if such drift is post-Columbian, the extant Native populations may not represent well the pre-Columbian Natives that admixed with European settlers. DNA from the remains of pre-Columbian Native Americans could shed light on the extent to which this might be the case.

A further complication is that some of the reference populations may have experienced admixture following the colonial period. For example, it is possible that the Iberian reference individuals examined here have less non-European (East/South Mediterranean and/or Sub-Saharan African) ancestry than individuals migrating to the Americas during the colonial period due to ongoing admixture with other Europeans. In this case SOURCEFIND may overestimate the contributions from the non-European groups. Because of this, estimates for each of the East/South /Mediterranean and African components should be interpreted as values over and above those present in the Spanish/Portuguese reference individuals examined. As noted above, the similarity in inferred dates for admixture involving East/South Mediterranean versus Iberian ancestry furthermore suggests that the individuals carrying this excess East/South Mediterranean ancestry migrated to Latin America during the colonial period."

2. In a couple of places (line 156, 202), ongoing admixture, or admixture over multiple generations, is invoked to explain more recent inferred dates of admixture. It would be useful to have a discussion of what the different signatures are expected to be of a single pulse of admixture v. admixture over multiple generations. For example, if admixture is ongoing, how does that skew estimates of admixture time inferred by GLOBETROTTER? Either a discussion with citations, or an illustration using simulations would be good.

This is a good point. It is challenging to distinguish between pulses and continuous admixture in many cases, particularly where the pulses have occurred near in time. Simulations and theoretical results from previous work using GLOBETROTTER suggest that in cases of either continuous admixture or multiple pulses of admixture involving the same sources, inferred (single) dates may be somewhere in between the start and end of admixture, occasionally with a bias towards more the more recent dates (Hellenthal et al 2014). However, admixture from the colonial era to the present-day spans a small enough time frame that the distinction between whether the admixture should be considered a “pulse” versus “continuous” seems subtle in this setting. For this reason, we use “continuing” admixture in lines 156 and 202 to reflect that it could be either in reality.

Regarding line 156, which involves a particular observation of how Native ancestry appears to be increasing over time towards the present-day in individuals with inferred Native+Iberian (and no other) ancestry, we used simulations to mimic and understand this effect, with these simulations (and interpretation) described in the sub-section “Simulations with two sequential admixture events” in Supplementary Note 1. We now refer the reader specifically to these simulations in this sentence in the main text:

“Noticeably, individuals with more recent inferred dates of admixture have greater Native ancestry (Fig. 3A, Supplementary Table 4), with simulations suggesting this is consistent with continuing admixture between admixed Latin Americans and unadmixed Natives (Supplementary Note 2)...”

Here is an excerpt from that Supplementary section, which we have edited for clarity:

“To further evaluate the trend of increasing Native ancestry at more recent dates of admixture seen in the CANDELA data, we simulated 1,050 additional individuals with two sequential admixture events. As before, we simulated different proportions of admixture from two sources (*CentralSouthSpain* and *Quechua2*), and varied the times for the two admixture events. Using the exponential sampling procedure described above, we first simulated individuals stemming from an instantaneous admixture event occurring 2 generations previously, with 55% *CentralSouthSpain* ancestry and 45% *Quechua2* ancestry. We then simulated a second instantaneous admixture event with p ancestry from the population generated in the first admixture event, and $1-p$ ancestry from *Quechua2* occurring g generations ago. We simulated $p = 0.86-0.98$ (at 0.02 intervals) and $g = 5-14$ generations, with 15 simulated individuals for each combination of p and g ($1,050$ simulated individuals in total). Note that, under this simulation procedure, the first admixture event occurred $g+2$ generations ago, the more recent event occurred g generations ago, and the final expected proportion of ancestry from *CentralSouthSpain* is $0.55 * p$. SOURCEFIND and GLOBETROTTER were run separately on each simulated individual as before. As with the previous section, for these simulations we used the more computationally efficient version of SOURCEFIND, described at the start of this Supplementary Note, to infer proportions.

In 923 ($\sim 88\%$) of the $1,050$ individuals, GLOBETROTTER concluded only a single date of admixture, which is not surprising given the inherent difficulty in distinguishing between two pulses of admixture separated by only 2 generations that involve the same source groups. The

figure below shows results when assuming a single date of admixture, which infers dates that typically are 2 generations above g (simulated date given with the grey bar). Therefore, GLOBETROTTER most often concludes a single date of admixture, with the inferred date primarily reflecting the older event because this is reflected in the sizes of observed Iberian ancestry segments.

In addition, as above, we extracted the 923 simulated individuals that GLOBETROTTER inferred to have a single admixture event between source groups that best-matched Native and European surrogate groups. We binned these individuals based on their inferred admixture date, and calculated the average ancestry inferred proportions in each bin. While not as striking as that observed in our real data (Fig. 3A of the main text), the figure below shows an analogous trend for decreasing Native American ancestry at increasing g that is significant ($p < 0.001$) under the same simple linear regression model used for analysing this trend in the real data (Supplementary Table 4). This is because individuals here are simulated with different proportions of admixture from the earlier admixture event occurring $g+2$ generations ago. Individuals with more simulated ancestry from this earlier admixed group have (i) more European ancestry and (ii) inferred dates that may be slightly older by retaining more signal from this older admixture event. Indeed, a simple linear regression of the bias in date estimate (in generations ago) for these 923 individuals on their expected proportion of Spanish ancestry shows a significantly positive association ($p < 0.007$). In contrast, for the 1,297 simulated individuals described in the previous section with only a single simulated admixture date, there is no such significant trend ($p = 0.33$). Overall these simulation results suggest that mixture between unadmixed and admixed Natives over time, such as that we simulated in this section, could lead to the trend we observe in Figure 3A.”

3. All of the Wilcoxon rank-sum tests are one-sided, but there is no discussion of why all other admixture events are expected to be more recent than the Iberian admixture. This should go somewhere.

Reviewer #3 has also raised this point, so we have added clarifications on this throughout the main text. In brief (see also comments to Reviewer #3), we use Iberia as a time reference in this analysis because Iberia is believed to represent the source of the first colonial-era migrants to Latin America (with our inferred dates supporting this view).

4. I found the explanation of the pipeline in the methods extremely confusing (the use of CHROMOPAINTER, NNLS, fineSTRUCTURE, and SOURCEFIND, and the definition of all of the different sets of population clusters) — a schematic figure (probably in the supplement) would be extremely useful here. In particular, it is a bit unclear why you use both NNLS and SOURCEFIND, and what they are doing differently. I gather that NNLS is used as a preliminary step because it is more computationally efficient? This should be made much more clear upfront to help orient the reader to what each step of the pipeline is doing.

We agree and have now included two schematic figures, one explaining all steps of the analyses and another explaining the particular steps used in deciding clusters in more detail, which used NNLS as the reviewer notes. NNLS was used here over SOURCEFIND not only for computational convenience, as the reviewer notes, but also for testing the ancestry estimation with the simplest regression model. We have added more information in the methods section:

“...We performed this analysis (and the further refinements described below) using NNLS (instead of SOURCEFIND; described below) so as to identify the surrogate clusters detected by the simplest

regression model (NNLS being a numerical optimization technique while SOURCEFIND is a model-based approach)...”

Other uses of NNLS in this paper were just used to compare accuracy relative to SOURCEFIND via simulations.

5. Ideally, the SOURCEFIND software should be made publicly available as a github repository (e.g.) to better facilitate reproduction of these results, as well as the furthering of other studies.

We agree, and have made SOURCEFIND available upon request from one of the authors (g.hellenthal@ucl.ac.uk), as stated in the Supplementary Material.

Specific Comments:

1. line 125: "simulations show that SOURCEFIND has greater accuracy than other approaches..." It would be helpful to name them here. By the supplement, it looks like the only comparison is to NNLS. Are there others that do similar tasks? If so, some brief discussion of those would be appropriate. Also, from what I understand, although it is more accurate, SOURCEFIND is more computationally expensive — it would be good to mention that here or elsewhere in the main text, especially in the context of the analysis pipeline.

The reviewer raises a very good point here as we failed to elaborate these explanations better, but we have now improved the text accordingly. The simulated data have only been analysed with NNLS and SOURCEFIND, so we have updated the above sentence:

“Simulations show that SOURCEFIND has greater accuracy than NNLS.”

We also include a brief note on SOURCEFIND’s computation time and the accuracy of two different applications of the SOURCEFIND methodology used in this paper. Specifically, while the version we applied to the CANDELA data was very computationally expensive, we also have a much less intensive version (which will be the released version) that gives similar accuracy without very severe computation restraints, as we now explicitly mention in the 2nd paragraph of Supplementary Note 2: “In particular in each of simulations (i)-(iv) described below, we provide plots illustrating the accuracy of both the initial SOURCEFIND version (called “SOURCEFIND1” in this section) and the computationally efficient version of SOURCEFIND (called “SOURCEFIND2”). For these simulations, accuracy is only very slightly reduced when using SOURCEFIND2 relative to SOURCEFIND1. Regarding computation time, analysis of a single CANDELA individual took ~10 minutes using each run of the initial SOURCEFIND version with 200K MCMC iterations, hence taking 10x50=500 minutes to do 50 independent runs. In contrast, it took ~25 seconds with 100K MCMC iterations and a single run of the more computationally efficient version. We note that additional independent runs of the computationally efficient version may improve performance, while reducing the gains in computation time (e.g. doing 50 independent runs would make it only ~20x faster than the initial SOURCEFIND).”

Furthermore, we note that analyses on the CANDELA data have shown the limitations of perhaps the most popular ancestry testing approach, ADMIXTURE, confirming findings from previous studies (Lawson 2012, Lawson 2017). We have added an additional sentence to this paragraph:

“Analysis of these data using the allele-frequency-based approach ADMIXTURE (Alexander et al. 2009) show major limitations for estimating sub-continental ancestry (Supplementary Note 1), similar to what is seen in other datasets (Lawson et al. 2017, Lawson et al. 2012). We therefore performed fully haplotype-based analyses.”

Moreover, we note that other approaches like PCA or AS-PCA are not directly comparable as it is not possible to draw percentages from them.

2. line 192: "arguably with a peculiar history" — what does this mean? I'm intrigued, but confused.

By “peculiar history” we mean certain populations for which there is some historical evidence of a converso/crypto-jewish contribution, of which there are a few across the Americas. Probably the most widely studied are certain Latino groups from the US South West (“To the end of the earth: a history of the crypto-jews of New Mexico”, S.M. Hordes 2005) and a few other isolated populations (e.g. Velez et al. Hum Genet (2012) 131:251–263). We have rephrased this section of the manuscript to clarify what we mean.

3. line 195-196: I'm sort of confused by this sentence. Would you expect Sub-Saharan ancestry to be higher than 4%? Why are regions that received large numbers of African slaves under-represented?

We have expanded this section to explain this better:

“It has been estimated that Brazil received about 4.2 million African slaves (about half of those brought to the Americas) while Spanish America received (altogether) about 1.5 million (Adhikari et al, 2017). However, the average Sub-Saharan ancestry in the full CANDELA sample is relatively low (<4%), probably reflecting the fact that regions which historically received large numbers of slaves are under-represented in this sample (particularly for Brazil, which was sampled mainly in the South) (Ruiz-Linares et al. 2014). Altogether, ~22% of the individuals studied show more than 5% sub-Saharan African ancestry.”

4. line 206-208: Does more Chinese ancestry in the non-Brazilian CANDELA individuals, and more Japanese ancestry in the Brazilian individuals match any migration documentation? Or could this be an artifact of the Brazilian individuals having different distributions of other ancestry components?

This section has been improved to reflect the historical evidences that could support these findings:

“Other than the major Native American, Caucasian and sub-Saharan African ancestry components, historical information indicates some East Asian migration to Latin America, particularly after independence in the 19th century¹⁹.”

“These results match historical records documenting the arrival of Chinese labourers to Peru since the middle 19th century³² and Japanese labourers to Brazil since the early 20th century³³.”

Reviewer #2 (Remarks to the Author):

Thank you for clearly stating and refuting the issues I have raised (reviewer #2). Apart from the original genetic structures are "mirrored", I am fine with all the rest of the answers and comments. I recommend it to be accepted and published.

The authors' main contribution is a very 'fine' mapping of the populations. This is significant an effort and result.

(Although, I wonder if it is just me not understanding it properly that "the structure is mirrored" is so important. Are you trying to say that Wang's 2008 paper emphasizes European influence, while there is much stronger native population conservation, genomically speaking?)

I think that "pre-Columbian structure is mirrored(?)" is a bit problematic to me. I guess the authors meant pre-Columbian genetic signatures are "strongly conserved"? If the structure is meant to indicate the ethnic portions(chart slice) are conserved, it is obvious.

On the HWE issue, the genetic "structures" are detected and shown by algorithms that use allele frequency statistics whether originally allele- or haplotype-based. What else?

The authors do not use SV, CNV, or TE kind structural variation data to really see the structural variations of the genomes as they use 500,000 SNPs.

So, if the genetic structure is 'mirrored', it means the allele frequencies are as expectedly conserved. Although in this case, there were admixtures to the existing populations.

As far as I know, all genetic/genomic structures, whether they are big or sub populations, are determined by some variant frequency statistics. Not going into details of the Bayes model, all such probability based methods should use allele frequencies of common and less common SNPs.

Haplotype/blocks are useful because they have variation signals from conserved allele frequencies. Even with admixtures at various times, the allele frequencies stay more or less the same, UNLESS there have been extreme or dramatic selections (whatever selection that would be) or some kind of sweeping. (And you cannot find those in these very close pops. The nose case is much older than Columbus)

If I put the question in another way, what other results would you expect/get if it was not the case (some kind of continuous allele frequency equilibrium)?

Did you expect that "pre-Columbian Native genetic structure/signature is NOT well-conserved in Latin Americans and that sources of non-Native ancestry, and admixture timings, DO NOT match documented migratory flows" due to some interesting natural, social, or particular endemic events or hypotheses?

Are there any on-going debates on the genetic variations or sub-structures of native, admixed, very recent, some hidden migratory events, etc, in those populations?

(my no novelty point was such conservation and overall genomic signatures were already known)

And the authors find clearly with your present algorithms and analyses that somehow (surprisingly?) the genetic structures have been indeed conserved.

>This reviewer thinks that the above is perhaps the most >important finding of the analyses. It is not surprising. Does >that just simply reflect HW equilibrium?

>We find this comment puzzling. Hardy Weinberg equilibrium

>(HWE) describes how allele

>frequencies remain constant in a population from generation to >generation, so it is unclear how

>this comment relates to the findings on Latin American genetic >structure that we report here.

* I thought, at the heart of the analyses, the allele frequencies (genetic signatures) remained more or less constant and hence the "genetic structures" remain the same. I.e., we know the pop history and we know the original pop genomes and we see the combination (admix.) of those. No unexpected deviation in the evolution of the populations, for centuries.

>Population sub-division and admixture are violations of the >assumptions of HWE, so our results – which strongly support >both – indicate that HWE is violated, though that may depend

>on how you define a "population". For such reasons, we feel >that commenting on HWE will confuse readers.

I agree. I do not suggest you put HWE. It seemed a very obvious result (structure/variants conservation) to me. Genetic structures are all conserved unless there are dramatic selection/events or some real structural variations (such as some long genomic regions are altered).

Perhaps the word 'mirrored' is difficult for me to understand.

(A son's genome is mirrored by his parents' genomes?)

====

The phenotypic case of nose suggests that there has been

some strong selection going on. It means there have been allele frequency deviations in contrasting sub populations.

(there are no post-Columbus geno-pheno selection?)

>morphological studies have extensively demonstrated that many >physical appearance traits vary between populations (see

>references in the paper) and in particular among populations >from the countries we examined here (Comas, J. Antropologia de >los pueblos iberoamericanos. Biblioteca Universitaria Labor,

>1974). The causes for this variation may be environmental

>(including social), as the reviewer proposes, but they >certainly exist.

>we demonstrate that variation in Native sub-continental >ancestry in the Andean region significantly impacts on facial >features such as nose morphology, and that variation in >Northern versus Southern European ancestry

>significantly impacts on pigmentation phenotypes among >Brazilians. We also provide additional >evidence by finding statistically significant differences in >allele frequencies between Central >Andeans and Mapuches in loci associated with variation in >facial traits.

>We have now included a new analysis that further shows the >allele frequencies at these loci are unusually differentiated

>between Central Andeans and Mapuches relative to genome-wide >average, suggesting they have been subjected to selection.

A good addition.

>Hopefully, the fact that this reviewer, a priori, did not >believe such associations should exist helps to showcase one >of the novelties of this paper.

>Furthermore, we find that ancestry related to highland

>(Central Andean) versus

>lowland (Mapuche) Natives is associated with variation in >facial features, particularly nose

>morphology, and detect significant differences in allele >frequencies between these groups at

>loci previously associated with nose morphology in this sample.

Yes, this confirmation is a valuable contribution.

Reviewer #3 (Remarks to the Author):

The authors were very responsive to my comments on the previous version of the manuscript. The revised manuscript is substantially improved and is more readable. My major concerns have been addressed.

I have a few more minor suggestions that would improve the paper:=

Figure 1: It would be good to make clearer in the caption that panels B-D refer to the entire CANDELA dataset and not the reference population samples.

Figure 3D label: Write out that “SSA” means Sub-Saharan African.

Data availability: A URL should be provided directing readers to where the GWAS summary statistics can be found.

Reviewer #4 (Remarks to the Author):

I am satisfied that my feedback, and that of the other reviewers, has been incorporated thoughtfully. I would still encourage the authors to release the software upfront (rather than requiring a specific request) since this would allow the community to use it more readily, and to post questions and issues publicly so that common issues (formatting of input files, small bugs, version issues, etc...) can be resolved efficiently. I do not think, however, that this should hold up publication of the manuscript.

REVIEWERS' COMMENTS:

REVIEWER #2

Thank you for clearly stating and refuting the issues I have raised (reviewer #2). Apart from the original genetic structures are "mirrored", I am fine with all the rest of the answers and comments. I recommend it to be accepted and published.

The authors' main contribution is a very 'fine' mapping of the populations. This is significant an effort and result.

(Although, I wonder if it is just me not understanding it properly that "the structure is mirrored" is so important. Are you trying to say that Wang's 2008 paper emphasizes European influence, while there is much stronger native population conservation, genomically speaking?)

I think that "pre-Columbian structure is mirrored(?)" is a bit problematic to me. I guess the authors meant pre-Columbian genetic signatures are "strongly conserved"? If the structure is meant to indicate the ethnic portions(chart slice) are conserved, it is obvious.

We apologize for the use of the word “mirrored”. We believe the reviewer is correct about what we were intending to say, which is that there seems to be geographical concordance between DNA patterns in current-day Native groups and those found in Latin Americans. We have changed this to: “Native American ancestry components in Latin Americans correspond geographically to the present-day genetic structure of Native groups”.

On the HWE issue, the genetic "structures" are detected and shown by algorithms that use allele frequency statistics whether originally allele- or haplotype-based. What else?

The authors do not use SV, CNV, or TE kind structural variation data to really see the structural variations of the genomes as they use 500,000 SNPs.

So, if the genetic structure is 'mirrored', it means the allele frequencies are as expectedly conserved. Although in this case, there were admixtures to the existing populations.

As far as I know, all genetic/genomic structures, whether they are big or sub populations, are determined by some variant frequency statistics. Not going into details of the Bayes model, all such probability based methods should use allele frequencies of common and less common SNPs.

Haplotype/blocks are useful because they have variation signals from conserved allele frequencies. Even with admixtures at various times, the allele frequencies stay more or less the same, UNLESS there have been extreme or dramatic selections (whatever selection that would be) or some kind of sweeping. (And you cannot find those in these very close pops. The nose case is much older than Columbus)

If I put the question in another way, what other results would you expect/get if it was not the case (some kind of continuous allele frequency equilibrium)?

Did you expect that "pre-Columbian Native genetic structure/signature is NOT well-conserved in Latin Americans and that sources of non-Native ancestry, and admixture timings, DO NOT match documented migratory flows" due to some interesting natural, social, or particular endemic events or hypotheses?

While the reviewer is broadly correct in saying that there appears to be a geographic continuity between pre-Columbian haplotype frequency to haplotype frequency in Native segments of admixed Latin Americans, we note this does not extend to every Native group. For example, Fig 2 shows that Latin Americans do not share many haplotypes with our Native reference groups Pima, Mixe, Nahua2 and Chaco2. This may be due to a number of reasons, in part the fact that we are using modern-day surrogates that may be imperfect representatives of the original contributing Native groups (e.g. some of which may not be extinct). In this sense, we find the strength of continuity fairly striking and indicative of how colonial settlers intermixed with local Native groups and that those groups still reside in the same regions. It also illustrates the accuracy of the new SOURCEFIND software we are releasing with this paper.

Are there any on-going debates on the genetic variations or sub-structures of native, admixed, very recent, some hidden migratory events, etc, in those populations? (my no novelty point was such conservation and overall genomic signatures were already known)

And the authors find clearly with your present algorithms and analyses that somehow(surprisingly?) the genetic structures have been indeed conserved.

As noted above, we do find a striking continuity, though of course also show that demographic processes such as drift and admixture have greatly altered the genetic landscape of these countries. Some Latin Americans show DNA contributions from multiple Native groups, which may reflect other migratory events among Natives, though we note it could also reflect a lack of samples from colonial era Native groups (i.e. aDNA), which is an important area of future research in this area.

> This reviewer thinks that the above is perhaps the most important finding of the analyses. It is not surprising. Does that just simply reflect HW equilibrium?

> We find this comment puzzling. Hardy Weinberg equilibrium (HWE) describes how allele frequencies remain constant in a population from generation to generation, so it is unclear how this comment relates to the findings on Latin American genetic structure that we report here.

* I thought, at the heart of the analyses, the allele frequencies (genetic signatures) remained more or less constant and hence the "genetic structures" remain the same. I.e., we know the pop history and we know the original pop genomes and we see the

combination(admix.) of those. No unexpected deviation in the evolution of the populations, for centuries.

We broadly agree with this, as noted above.

> Population sub-division and admixture are violations of the assumptions of HWE, so our results – which strongly support both – indicate that HWE is violated, though that may depend on how you define a "population". For such reasons, we feel that commenting on HWE will confuse readers.

I agree. I do not suggest you put HWE. It seemed a very obvious result (structure/variants conservation) to me. Genetic structures are all conserved unless there are dramatic selection/events or some real structural variations (such as some long genomic regions are altered).

Our response to the above question also includes an explanation on this.

Perhaps the word 'mirrored' is difficult for me to understand.
(A son's genome is mirrored by his parents' genomes?)

We have reworded the abstract and removed the word 'mirrored'.

====

The phenotypic case of nose suggests that there has been some strong selection going on. It means there have been allele frequency deviations in contrasting sub populations.
(there are no post-Columbus geno-pheno selection?)

The reviewer is correct in summarizing this. We argue in the paper that there are opposing directions of selection in the Native populations, possibly due to their contrasting environments.

> morphological studies have extensively demonstrated that many physical appearance traits vary between populations (see references in the paper) and in particular among populations from the countries we examined here (Comas, J. Antropologia de los pueblos iberoamericanos. Biblioteca Universitaria Labor, 1974). The causes for this variation may be environmental (including social), as the reviewer proposes, but they certainly exist.

> we demonstrate that variation in Native sub-continental ancestry in the Andean region significantly impacts on facial >features such as nose morphology, and that variation in Northern versus Southern European ancestry significantly impacts on pigmentation phenotypes among >Brazilians. We also provide additional evidence by finding statistically significant differences in >allele frequencies between Central Andeans and Mapuches in loci associated with variation in facial traits.

We have now included a new analysis that further shows the allele frequencies at these loci are unusually differentiated between Central Andeans and Mapuches relative to genome-wide average, suggesting they have been subjected to selection.

A good addition.

Thank you.

> Hopefully, the fact that this reviewer, a priori, did not believe such associations should exist helps to showcase one >of the novelties of this paper.

> Furthermore, we find that ancestry related to highland (Central Andean) versus >lowland (Mapuche) Natives is associated with variation in facial features, particularly nose morphology, and detect significant differences in allele frequencies between these groups at loci previously associated with nose morphology in this sample.

Yes, this confirmation is a valuable contribution.

Thank you.

REVIEWER #3

The authors were very responsive to my comments on the previous version of the manuscript. The revised manuscript is substantially improved and is more readable. My major concerns have been addressed.

I have a few more minor suggestions that would improve the paper:=-

Figure 1: It would be good to make clearer in the caption that panels B-D refer to the entire CANDELA dataset and not the reference population samples.

Thank you for spotting this. We have added this to the caption: Panels (B), (C) and (D) refer to the CANDELA dataset.

Figure 3D label: Write out that "SSA" means Sub-Saharan African.

Thank you for the suggestion. We have replaced SSA in the figure for Sub-Saharan Africa, which also is in line with all the other panel titles.

Data availability: A URL should be provided directing readers to where the GWAS summary statistics can be found.

A URL has been provided as suggested by the reviewer.

REVIEWER #4

I am satisfied that my feedback, and that of the other reviewers, has been incorporated thoughtfully. I would still encourage the authors to release the software upfront (rather than requiring a specific request) since this would allow the community to use it more readily, and to post questions and issues publicly so that common issues (formatting of input files, small bugs, version issues, etc...) can be resolved efficiently. I do not think, however, that this should hold up publication of the manuscript.

We appreciate the reviewer's suggestion and have uploaded SOURCEFIND to www.paintmychromosomes.com.